# Novelty Detection via Robust Variational Autoencoding

## Abstract

We propose a new method for novelty detection that can tolerate high corruption of the training points, whereas previous works assumed either no or very low corruption. Our method trains a robust variational autoencoder (VAE), which aims to generate a model for the uncorrupted training points. To gain robustness to high corruption, we incorporate the following four changes to the common VAE: 1. Extracting crucial features of the latent code by a carefully designed dimension reduction component for distributions; 2. Modeling the latent distribution as a mixture of Gaussian low-rank inliers and full-rank outliers, where the testing only uses the inlier model; 3. Applying the Wasserstein-1 metric for regularization, instead of the Kullback-Leibler (KL) divergence; and 4. Using a least absolute deviation error for reconstruction. We establish both robustness to outliers and suitability to low-rank modeling of the Wasserstein metric as opposed to the KL divergence. We illustrate state-of-the-art results on standard benchmarks for novelty detection.

## 1 Introduction

Novelty detection refers to the task of detecting testing data points that deviate from the underlying structure of a given training dataset (Chandola et al., 2009; Pimentel et al., 2014; Chalapathy & Chawla, 2019). It finds crucial applications, in areas such as insurance and credit fraud (Zhou et al., 2018), mobile robots (Neto & Nehmzow, 2007) and medical diagnosis (Wei et al., 2018). Ideally, novelty detection requires learning the underlying distribution of the training data, where sometimes it is sufficient to learn a significant feature, geometric structure or another property of the training data. One can then apply the learned distribution (or property) to detect deviating points in the test data. This is different from outlier detection (Chandola et al., 2009), in which one does not have training data and has to determine the deviating points in a sufficiently large dataset assuming that the majority of points share the same structure or properties. We note that novelty detection is equivalent to the well-known one-class classification problem (Moya & Hush, 1996). In this problem, one needs to identify members of a class in a test dataset, and consequently distinguish them from "novel" data points, given training points from this class. The points of the main class are commonly referred to as inliers and the novel ones as outliers.

Novelty detection is also commonly referred to as semi-supervised anomaly detection. In this terminology, the notion of being "semi-supervised" is different than usual. It emphasizes that only the inliers are trained, where there is no restriction on the fraction of training points. On the other hand, the unsupervised case has no training (we referred to this setting above as "outlier detection") and in the supervised case there are training datasets for both the inliers and outliers. We remark that some authors refer to semi-supervised anomaly detection as the setting where a small amount of labeled data is provided for both the inliers and outliers (Ruff et al., 2020).

There are a myriad of solutions to novelty detection. Nevertheless, such solutions often assume that the training set is purely sampled from a single class or that it has a very low fraction of corrupted samples. This assumption is only valid when the area of investigation has been carefully studied and there are sufficiently precise tools to collect data. However, there are different important scenarios, where this assumption does not hold. One scenario includes new areas of studies, where it is unclear how to distinguish between normal and abnormal points. For example, in the beginning of the COVID-19 pandemic it was hard to diagnose COVID-19 patients and distinguish them from other patients with pneumonia. Another scenario occurs when it is very hard to make precise measurements, for example, when working with the highly corrupted images obtained in cryogenic electron microscopy (cryo-EM).

Therefore, we study a robust version of novelty detection that allows a nontrivial fraction of corrupted samples, namely outliers, within the training set. We solve this problem by using a special variational

autoencoder (VAE) (Kingma & Welling, 2014). Our VAE is able to model the underlying distribution of the uncorrupted data, despite nontrivial corruption. We refer to our new method as "Mixture Autoencoding with Wasserstein penalty", or "MAW". In order to clarify it, we first review previous works and then explain our contributions in view of these works.

## 1.1 PREVIOUS WORK

Solutions to one-class classification and novelty detection either estimate the density of the inlier distribution (Bengio & Monperrus, 2005; Ilonen et al., 2006) or determine a geometric property of the inliers, such as their boundary set (Breunig et al., 2000; Schölkopf et al., 2000; Xiao et al., 2016; Wang & Lan, 2020; Jiang et al., 2019). When the inlier distribution is nicely approximated by a low-dimensional linear subspace, Shyu et al. (2003) proposes to distinguish between inliers and outliers via Principal Component Analysis (PCA). In order to consider more general cases of nonlinear low-dimensional structures, one may use autoencoders (or restricted Boltzmann machines), which nonlinearly generalize PCA (Goodfellow et al., 2016, Ch. 2) and whose reconstruction error naturally provides a score for membership in the inlier class. Instances of this strategy with various architectures include Zhai et al. (2016); Zong et al. (2018); Sabokrou et al. (2018); Perera et al. (2019); Pidhorskyi et al. (2018). In all of these works, but Zong et al. (2018), the training set is assumed to solely represent the inlier class. In fact, Perera et al. (2019) observed that interpolation of a latent space, which was trained using digit images of a complex shape, can lead to digit representation of a simple shape. If there are also outliers (with a simple shape) among the inliers (with a complex shape), encoding the inlier distribution becomes even more difficult. Nevertheless, some previous works already explored the possibility of corrupted training set (Xiao et al., 2016; Wang & Lan, 2020; Zong et al., 2018). In particular, Xiao et al. (2016); Zong et al. (2018) test artificial instances with at most $5\%$ corruption of the training set and Wang & Lan (2020) considers ratios of $10\%$, but with very small numbers of training points. In this work we consider corruption ratios up to $30\%$, with a method that tries to estimate the distribution of the training set, and not just a geometric property.

VAEs (Kingma & Welling, 2014) have been commonly used for generating distributions with reconstruction scores and are thus natural for novelty detection without corruption. They determine the latent code of an autoencoder via variational inference (Jordan et al., 1999; Blei et al., 2017). Alternatively, they can be viewed as autoencoders for distributions that penalize the Kullback-Leibler (KL) divergence of the latent distribution from the prior distribution. The first VAE-based method for novelty detection was suggested by An & Cho (2015). It was recently extended by Daniel et al. (2019) who modified the training objective. A variety of VAE models were also proposed for special anomaly detection problems, which are different than novelty detection (Xu et al., 2018; Zhang et al., 2019; Pol et al., 2019). Current VAE-based methods for novelty detection do not perform well when the training data is corrupted. Indeed, the learned distribution of any such method also represents the corruption, that is, the outlier component. To the best of our knowledge, no effective solutions were proposed for collapsing the outlier mode so that the trained VAE would only represent the inlier distribution.

An adversarial autoencoder (AAE) (Makhzani et al., 2016) and a Wasserstein autoencoder (WAE) (Tolstikhin et al., 2018) can be considered as variants of VAE. The penalty term of AAE takes the form of a generative adversarial network (GAN) (Goodfellow et al., 2016), where its generator is the encoder. A Wasserstein autoencoder (WAE) (Tolstikhin et al., 2018) generalizes AAE with a framework that minimizes the Wasserstein metric between the sample distribution and the inference distribution. It reformulates the corresponding objective function so that it can be implemented in the form of an AAE.

There are two relevant lines of works on robustness to outliers in linear modeling that can be used in nonlinear settings via autoencoders or VAEs. Robust PCA aims to deal with sparse elementwise corruption of a data matrix (Candès et al., 2011; De La Torre & Black, 2003; Wright et al., 2009; Vaswani & Narayanamurthy, 2018). Robust subspace recovery (RSR) aims to address general corruption of selected data points and thus better fits the framework of outliers (Watson, 2001; De La Torre & Black, 2003; Ding et al., 2006; Zhang et al., 2009; McCoy & Tropp, 2011; Xu et al., 2012; Lerman & Zhang, 2014; Zhang & Lerman, 2014; Lerman et al., 2015; Lerman & Maunu, 2017; Maunu et al., 2019; Lerman & Maunu, 2018; Maunu & Lerman, 2019). Autoencoders that use robust PCA for anomaly detection tasks were proposed in Chalapathy et al. (2017); Zhou & Paffenroth (2017). Dai et al. (2018) show that a VAE can be interpreted as a nonlinear robust PCA problem. Nevertheless, explicit regularization is often required to improve robustness to sparse corruption in VAEs (Akrami et al., 2019; Eduardo et al., 2020). RSR was successfully applied to outlier detection by Lai et al. (2020). One can apply their work to the different setting of novelty detection; however, our proposed VAE formulation seems to work better.

### 1.2 THIS WORK

We propose a robust novelty detection procedure, MAW, that aims to model the distribution of the training data in the presence of nontrivial fraction of outliers. We highlight its following four features:

- MAW models the latent distribution by a Gaussian mixture of low-rank inliers and full-rank outliers, and applies the inlier distribution for testing. Previous applications of mixture models for novelty detection were designed for multiple modes of inliers and used more complicated tools such as constructing another network (Zong et al., 2018) or applying clustering (Aytekin et al., 2018; Lee et al., 2018).
- MAW applies a novel dimension reduction component, which extracts lower-dimensional features of the latent distribution. The reduced small dimension allows using full covariances for both the outliers (with full rank) and inliers (with deficient rank); whereas previous VAE-based methods for novelty detection used diagonal covariances in their models (An & Cho, 2015; Daniel et al., 2019). The new component is inspired by the RSR layer in Lai et al. (2020); however, they are essentially different since the RSR layer is only applicable for data points and not for probability distributions.
- For the latent code penalty, MAW uses the Wasserstein-1 ($W_1$) metric. Under a special setting, we prove that the Wasserstein metric gives rise to outliers-robust estimation and is suitable to the low-rank modeling of inliers by MAW. We also show that these properties do not hold for the KL divergence, which is used by VAE, AAE and WAE. We remark that the use of the Wasserstein metric in WAE is different than that of MAW. Indeed, in WAE it measures the distance between the data distribution and the generated distribution and it does not appear in the latent code. Our use of $W_1$ can be viewed as a variant of AAE, which replaces GAN with Wasserstein GAN (WGAN) (Arjovsky et al., 2017). That is, it replaces the minimization of the KL divergence by that of the $W_1$ distance.
- MAW achieves state-of-the-art results on popular anomaly detection datasets.

Additional two features are as follows. First, for reconstruction, MAW replaces the common least squares formulation with a least absolute deviations formulation. This can be justified by the use of either a robust estimator (Lopuhaä & Rousseeuw, 1991) or a likelihood function with a heavier tail. Second, MAW is attractive for practitioners. It is simple to implement in any standard deep learning library, and is easily adaptable to other choices of network architecture, energy functions and similarity scores.

We remark that since we do not have labels for the training set, we cannot supervisedly learn the Gaussian component with low-rank covariance by the inliers and Gaussian component with the full-rank covariance by the outliers. However, the use of two robust losses (least absolute deviation and the $W_1$ distance) helps obtain a careful model for the inliers, which is robust to outliers. Note that in our testing, we only use the model for the inliers.

We explain MAW in §2. We establish the advantage of its use of the Wasserstein metric in §3. We carefully test MAW in §4. At last, we conclude this work in §5.

## 2 DESCRIPTION OF MAW

We motivate and overview the underlying model and assumptions of MAW in §2.1. We describe the simple implementation details of its components in §2.2. Fig. 1 illustrates the general idea of MAW and can assist in reading this section.

### 2.1 THE MODEL AND ASSUMPTIONS OF MAW

MAW aims to robustly estimate a mixture inlier-outlier distribution for the training data and then use its inlier component to detect outliers in the testing data. For this purpose, it designs a novel variational autoencoder with an underlying mixture model and a robust loss function in the latent space. We find the variational framework natural for novelty detection. Indeed, it learns a distribution that describes the inlier training examples and generalizes to the inlier test data. Moreover, the variational formulation allows a direct modeling of a Gaussian mixture model in the latent space, unlike a standard autoencoder.

We assume $L$ training points in $\mathbb{R}^D$, which we designate by $\{\mathbf{x}^{(i)}\}_{i=1}^L$. Let $\mathbf{x}$ be a random variable on $\mathbb{R}^D$ with the unknown training data distribution that we estimate by the empirical distribution of the training points. We assume a latent random variable $\mathbf{z}$ of low and even dimension $2 \leq d \leq D$, where our default choice is $d = 2$. We further assume a standardized Gaussian prior, $p(\mathbf{z})$, so that $\mathbf{z} \sim \mathcal{N}(\mathbf{0}, \boldsymbol{I}_{d \times d})$. The posterior distribution $p(\mathbf{z}|\mathbf{x})$ is unknown. However, we assume an approximation to it, which we denote by $q(\mathbf{z}|\mathbf{x})$, such that $\mathbf{z}|\mathbf{x}$ is a mixture of two Gaussian distributions representing the inlier and outlier components. More specifically, $\mathbf{z}|\mathbf{x} \sim \eta \mathcal{N}(\boldsymbol{\mu}_1, \boldsymbol{\Sigma}_1) + (1 - \eta)\mathcal{N}(\boldsymbol{\mu}_2, \boldsymbol{\Sigma}_2)$, where we explain next its parameters. We assume that $\eta > 0.5$, where our default value is $\eta = 5/6$, so that the

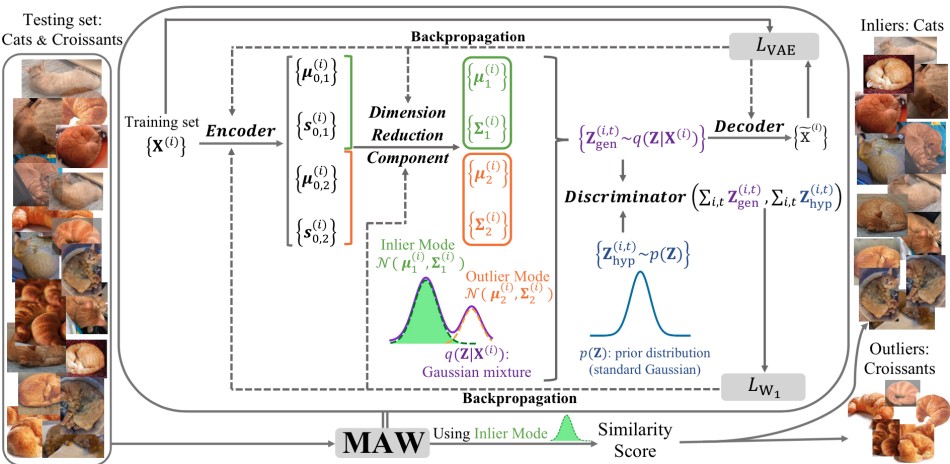

Figure 1: Demonstration of the architecture of MAW for novelty detection.

first mode of $\mathbf{z}$ represents the inliers and the second one represents the outliers. The other parameters are generated by the encoder network and a following dimension reduction component. We remark that unlike previous works which adopted Gaussian mixtures to model the clusters of inliers (Reddy et al., 2017; Zong et al., 2018), the Gaussian mixture model in MAW aims to separate between inliers and outliers. The dimension reduction component involves a mapping from a higher-dimensional space onto the latent space. It is analogous to the RSR layer in Lai et al. (2020) that projects encoded points onto the latent space, but requires a more careful design since we consider a distribution rather than sample points. Due to this reduction, we assume that the mapped covariance matrices of $\mathbf{z}|\mathbf{x}$ are full, unlike common single-mode VAE models that assume a diagonal covariance (Kingma & Welling, 2014; An & Cho, 2015). Our underlying assumption is that the inliers lie on a low-dimensional structure and we thus enforce the lower rank $d/2$ for $\mathbf{\Sigma}_1$, but allow $\mathbf{\Sigma}_2$ to have full rank $d$. Nevertheless, we later describe a necessary regularization of both matrices by the identity.

Following the VAE framework, we approximate the unknown posterior distribution $p(\mathbf{z}|\mathbf{x})$ within the variational family $\mathcal{Q} = \{q(\mathbf{z}|\mathbf{x})\}$, which is indexed by $\boldsymbol{\mu}_1$, $\mathbf{\Sigma}_1$, $\boldsymbol{\mu}_2$ and $\mathbf{\Sigma}_2$. Unlike a standard VAE, which maximizes the evidence lower bound (ELBO), MAW maximizes the following ELBO-Wasserstein, or ELBOW, function, which uses the $W_1$ distance (see also §A.1):
$$\text{ELBOW}(q) = \mathbb{E}_{p(\mathbf{x})}\mathbb{E}_{q(\mathbf{z}|\mathbf{x})} \log p(\mathbf{x}|\mathbf{z}) - W_1(q(\mathbf{z}), p(\mathbf{z})) . \tag{1}$$

Following the VAE framework, we use a Monte-Carlo approximation to estimate $\mathbb{E}_{q(\mathbf{z}|\mathbf{x})} \log p(\mathbf{x}|\mathbf{z})$ with i.i.d. samples, $\{\mathbf{z}^{(t)}\}_{t=1}^T$, from $q(\mathbf{z}|\mathbf{x})$ as follows:
$$\mathbb{E}_{q(\mathbf{z}|\mathbf{x})} \log p(\mathbf{x}|\mathbf{z}) \approx \frac{1}{T} \sum_{t=1}^T \log p(\mathbf{x}|\mathbf{z}^{(t)}). \tag{2}$$

To improve the robustness of our model, we choose the negative log likelihood function $-\log p(\mathbf{x}|\mathbf{z}^{(t)})$ to be a constant multiple of the $\ell_2$ norm of the difference of the random variable $\mathbf{x}$ and a mapping of the sample $\mathbf{z}^{(t)}$ from $\mathbb{R}^d$ to $\mathbb{R}^D$ by the decoder, $\mathcal{D}$, that is,
$$-\log p(\mathbf{x}|\mathbf{z}^{(t)}) \propto \left\| \mathbf{x} - \mathcal{D}(\mathbf{z}^{(t)}) \right\|_2 . \tag{3}$$
Note that we deviate from the common choice of the squared $\ell_2$ norm, which corresponds to an underlying Gaussian likelihood and assume instead a likelihood with a heavier tail.

MAW trains its networks by minimizing –ELBOW($q$). For any $1 \le i \le L$, it samples $\{\mathbf{z}_{\text{gen}}^{(i,t)}\}_{t=1}^T$ from $q(\mathbf{z}|\mathbf{x}^{(i)})$, where all samples are independent. Using the aggregation formula: $q(\mathbf{z}) = L^{-1} \sum_{i=1}^L q(\mathbf{z}|\mathbf{x}^{(i)})$, which is also used by an AAE, the approximation of $p(\mathbf{x})$ by the empirical distribution of the training data, and (1)-(3), MAW applies the following approximation of –ELBOW($q$):
$$-\frac{1}{LT} \sum_{i=1}^L \sum_{t=1}^T \left\| \mathbf{x}^{(i)} - \mathcal{D}(\mathbf{z}_{\text{gen}}^{(i,t)}) \right\|_2 + W_1 \left( \frac{1}{L} \sum_{i=1}^L q(\mathbf{z}|\mathbf{x}^{(i)}), p(\mathbf{z}) \right) . \tag{4}$$
Details of minimizing (4) are described in §2.2. We remark that the procedure described in §2.2 is independent of the multiplicative constant in (3) and therefore this constant is ignored in (4).

During testing, MAW identifies inliers and outliers according to high or low similarity scores computed between each given test point and points generated from the learned inlier component of $\mathbf{z}|\mathbf{x}$.

## 2.2 DETAILS OF IMPLEMENTING MAW

MAW has a VAE-type structure with additional WGAN-type structure for minimizing the $W_1$ loss in (4). We provide here details of implementing these structures. Some specific choices of the networks are described in §4 since they may depend on the type of datasets.

The VAE-type structure of MAW contains three ingredients: encoder, dimension reduction component and decoder. The encoder forms a neural network $\mathcal{E}$ that maps the training sample $\mathbf{x}^{(i)} \in \mathbb{R}^D$ to $\boldsymbol{\mu}_{0,1}^{(i)}, \boldsymbol{\mu}_{0,2}^{(i)}, \boldsymbol{s}_{0,1}^{(i)}, \boldsymbol{s}_{0,2}^{(i)}$ in $\mathbb{R}^{D'}$, where our default choice is $D' = 128$. The dimension reduction component then computes the following statistical quantities of the Gaussian mixture $\mathbf{z}|\mathbf{x}^{(i)}$: means $\boldsymbol{\mu}_1^{(i)}$ and $\boldsymbol{\mu}_2^{(i)}$ in $\mathbb{R}^d$ and covariance matrices $\boldsymbol{\Sigma}_1^{(i)}$ and $\boldsymbol{\Sigma}_2^{(i)}$ in $\mathbb{R}^{d \times d}$. First, a linear layer, represented by $\boldsymbol{A} \in \mathbb{R}^{D' \times d}$, maps (via $\boldsymbol{A}^\mathrm{T}$) the features $\boldsymbol{\mu}_{0,1}^{(i)}$ and $\boldsymbol{\mu}_{0,2}^{(i)}$ in $\mathbb{R}^{D'}$ to the following respective vectors in $\mathbb{R}^d$: $\boldsymbol{\mu}_1^{(i)} = \boldsymbol{A}^\mathrm{T}\boldsymbol{\mu}_{0,1}^{(i)}$ and $\boldsymbol{\mu}_2^{(i)} = \boldsymbol{A}^\mathrm{T}\boldsymbol{\mu}_{0,2}^{(i)}$. For $j = 1, 2$, form $\boldsymbol{M}_j^{(i)} = \boldsymbol{A}^\mathrm{T}\mathrm{diag}(\boldsymbol{s}_{0,j}^{(i)})\boldsymbol{A}$. For $j = 2$, compute $\boldsymbol{\Sigma}_2^{(i)} = \boldsymbol{M}_2^{(i)}\boldsymbol{M}_2^{(i)\mathrm{T}}$. For $j = 1$, we first need to reduce the rank of $\boldsymbol{M}_1^{(i)}$. For this purpose, we form

$$\boldsymbol{M}_1^{(i)} = \boldsymbol{U}_1^{(i)}\mathrm{diag}(\boldsymbol{\sigma}_1^{(i)})\boldsymbol{U}_1^{(i)\mathrm{T}}, \tag{5}$$

the spectral decomposition of $\boldsymbol{M}_1^{(i)}$, and then truncate its bottom $d/2$ eigenvalues. That is, let $\tilde{\boldsymbol{\sigma}}_1^{(i)} \in \mathbb{R}^d$ have the same entries as the largest $d/2$ entries of $\boldsymbol{\sigma}_1^{(i)}$ and zero entries otherwise. Then, compute

$$\tilde{\boldsymbol{M}}_1^{(i)} = \boldsymbol{U}_1^{(i)\mathrm{T}}\mathrm{diag}(\tilde{\boldsymbol{\sigma}}_1^{(i)})\boldsymbol{U}_1^{(i)} \tag{6}$$

and $\boldsymbol{\Sigma}_1^{(i)} = \tilde{\boldsymbol{M}}_1^{(i)}\tilde{\boldsymbol{M}}_1^{(i)\mathrm{T}}$. Since the TensorFlow package requires numerically-significant positive definiteness of covariance matrices, we add an identity matrix to both $\boldsymbol{\Sigma}_1^{(i)}$ and $\boldsymbol{\Sigma}_2^{(i)}$. Despite this, the low-rank structure of $\boldsymbol{\Sigma}_1^{(i)}$ is still evident. Note that the dimension reduction component only trains $\boldsymbol{A}$. The decoder, $\mathcal{D} : \mathbb{R}^d \to \mathbb{R}^D$, maps independent samples, $\{\mathbf{z}_{\mathrm{gen}}^{(i,t)}\}_{t=1}^T$, generated for each $1 \le i \le L$ by the distribution $\eta\mathcal{N}(\boldsymbol{\mu}_1^{(i)}, \boldsymbol{\Sigma}_1^{(i)}) + (1 - \eta)\mathcal{N}(\boldsymbol{\mu}_2^{(i)}, \boldsymbol{\Sigma}_2^{(i)})$, into the reconstructed data space.

The loss function associated with the VAE structure is the first term in (4). We can write it as

$$L_{\mathrm{VAE}}(\mathcal{E}, \boldsymbol{A}, \mathcal{D}) = \frac{1}{LT}\sum_{i=1}^L \sum_{t=1}^T \left\| \mathbf{x}^{(i)} - \mathcal{D}(\mathbf{z}_{\mathrm{gen}}^{(i,t)}) \right\|_2 . \tag{7}$$

The dependence of this loss on $\mathcal{E}$ and $\boldsymbol{A}$ is implicit, but follows from the fact that the parameters of the sampling distribution of each $\mathbf{z}_{\mathrm{gen}}^{(i,t)}$ were obtained by $\mathcal{E}$ and $\boldsymbol{A}$.

The WGAN-type structure seeks to minimize the second term in (4) using the dual formulation

$$W_1\left(\frac{1}{L}\sum_{i=1}^L q(\mathbf{z}|\mathbf{x}^{(i)}), p(\mathbf{z})\right) = \sup_{\|f\|_{Lip} \le 1} \mathbb{E}_{\mathbf{z}_{\mathrm{hyp}} \sim p(\mathbf{z})} f(\mathbf{z}_{\mathrm{hyp}}) - \mathbb{E}_{\mathbf{z}_{\mathrm{gen}} \sim \frac{1}{L}\sum_{i=1}^L q(\mathbf{z}|\mathbf{x}^{(i)})} f(\mathbf{z}_{\mathrm{gen}}). \tag{8}$$

The generator of this WGAN-type structure is composed of the encoder $\mathcal{E}$ and the dimension reduction component, which we represent by $\boldsymbol{A}$. It generates the samples $\{\mathbf{z}_{\mathrm{gen}}^{(i,t)}\}_{i=1,t=1}^{L,T}$ described above. The discriminator, $\mathcal{D}is$, of the WGAN-type structure plays the role of the Lipschitz function $f$ in (8). It compares the latter samples with the i.i.d. samples $\{\mathbf{z}_{\mathrm{hyp}}^{(i,t)}\}_{t=1}^T$ from the prior distribution. In order to make $\mathcal{D}is$ Lipschitz, its weights are clipped to $[-1, 1]$ during training. In the MinMax game of this WGAN-type structure, the discriminator minimizes and the generator ($\mathcal{E}$ and $\boldsymbol{A}$) maximizes

$$L_{W_1}(\mathcal{D}is) = \frac{1}{LT}\sum_{i=1}^L \sum_{t=1}^T \left( \mathcal{D}is(\mathbf{z}_{\mathrm{gen}}^{(i,t)}) - \mathcal{D}is(\mathbf{z}_{\mathrm{hyp}}^{(i,t)}) \right) . \tag{9}$$

We note that maximization of (9) by the generator is equivalent to minimization of the loss function

$$L_{\mathrm{GEN}}(\mathcal{E}, \boldsymbol{A}) = -\frac{1}{LT}\sum_{i=1}^L \sum_{t=1}^T \mathcal{D}is(\mathbf{z}_{\mathrm{gen}}^{(i,t)}) . \tag{10}$$

During the training phase, MAW alternatively minimizes the losses (7)-(10) instead of minimizing a weighted sum. Therefore, any multiplicative constant in front of either term of (4) will not effect the optimization. In particular, it was okay to omit the multiplicative constant of (3) when deriving (4).

For each testing point $\mathbf{y}^{(j)}$, we sample $\{\mathbf{z}_{\text{in}}^{(j,t)}\}_{t=1}^T$ from the inlier mode of the learned latent Gaussian mixture and decode them as $\{\tilde{\mathbf{y}}^{(j,t)}\}_{t=1}^T = \{\mathcal{D}(\mathbf{z}_{\text{in}}^{(j,t)})\}_{t=1}^T$. Using a similarity measure $S(\cdot, \cdot)$ (our default is the cosine similarity), we compute $S^{(j)} = \sum_{t=1}^T S(\mathbf{y}^{(j)}, \tilde{\mathbf{y}}^{(j,t)})$. If $S^{(j)}$ is larger than a chosen threshold, then $\mathbf{y}^{(j)}$ is classified normal, and otherwise, novel. Additional details of MAW are in §A.

## 3 THEORETICAL GUARANTEES FOR THE $W_1$ MINIMIZATION

Here and in §D we theoretically establish the superiority of using the $W_1$ distance over the KL divergence. We formulate a simplified setting that aims to isolate the minimization of the WGAN-type structure introduced in §2.2, while ignoring unnecessary complex components of MAW. We assume a mixture parameter $\eta > 1/2$, a separation parameter $\epsilon > 0$ and denote by $\mathcal{R}$ the regularizing function, which can be either the KL divergence or $W_1$, and by $\mathcal{S}_+^K$ and $\mathcal{S}_{++}^K$ the sets of $K \times K$ positive semidefinite and positive definite matrices, respectively. For $\mu_0 \in \mathbb{R}^K$ and $\Sigma_0 \in \mathcal{S}_{++}^K$, we consider the minimization problem

$$\min_{\substack{\mu_1, \mu_2 \in \mathbb{R}^K; \Sigma_1, \Sigma_2 \in \mathcal{S}_+^K \\ \text{s.t. } \|\mu_1 - \mu_2\|_2 \geq \epsilon}} \eta \mathcal{R}\left(\mathcal{N}(\mu_1, \Sigma_1), \mathcal{N}(\mu_0, \Sigma_0)\right) + (1 - \eta)\mathcal{R}\left(\mathcal{N}(\mu_2, \Sigma_2), \mathcal{N}(\mu_0, \Sigma_0)\right). \quad (11)$$

We further motivate it in §D.1. For MAW, $\mu_0 = \mathbf{0}$ and $\Sigma_0 = \mathbf{I}$, but our generalization helps clarify things. This minimization aims to approximate the "prior" distribution $\mathcal{N}(\mu_0, \Sigma_0)$ with a Gaussian mixture distribution. The constraint $\|\mu_1 - \mu_2\|_2 \geq \epsilon$ distinguishes between the inlier and outlier modes and it is a realistic assumption as long as $\epsilon$ is sufficiently small.

Our cleanest result is when $\Sigma_0$, $\Sigma_1$ and $\Sigma_2$ coincide. It demonstrates robustness to the outlier component by the $W_1$ (or $W_p$, $p \geq 1$) minimization and not by the KL minimization (its proof is in §D.2).

**Proposition 3.1.** *If $\mu_0 \in \mathbb{R}^K$, $\Sigma_0 \in \mathcal{S}_{++}^K$, $\epsilon > 0$ and $1 > \eta > 1/2$, then the minimizer of (11) with $\mathcal{R} = W_p$, $p \geq 1$ and the additional constraint: $\Sigma_0 = \Sigma_1 = \Sigma_2$, satisfies $\mu_1 = \mu_0$, and thus the recovered inlier distribution coincides with the "prior distribution". However, the minimizer of (11) with $\mathcal{R} = KL$ and the same constraint satisfies $\mu_0 = \eta\mu_1 + (1 - \eta)\mu_2$.*

In §D.3, we analyze the case where $\Sigma_1$ is low rank and $\Sigma_2 \in \mathcal{S}_{++}^K$. We show that (11) is ill-defined when $\mathcal{R} = KL$. The $\mathcal{R} = W_1$ case is hard to analyze, but we can fully analyze the $\mathcal{R} = W_2$ case and demonstrate exact recovery of the prior distribution by the inlier distribution when $\eta$ approaches 1.

## 4 EXPERIMENTS

We describe the competing methods and experimental choices in §4.1. We report on the comparison with the competing methods in §4.2. We demonstrate the importance of the novel features of MAW in §4.3.

### 4.1 COMPETING METHODS AND EXPERIMENTAL CHOICES

We compared MAW with the following methods (descriptions and code links are in §E): Deep Autoencoding Gaussian Mixture Model (DAGMM) (Zong et al., 2018), Deep Structured Energy-Based Models (DSEBMs) (Zhai et al., 2016), Isolation Forest (IF) (Liu et al., 2008), Local Outlier Factor (LOF) (Breunig et al., 2000), One-class Novelty Detection Using GANs (OCGAN) (Perera et al., 2019), One-Class SVM (OCSVM) (Heller et al., 2003) and RSR Autoencoder (RSRAE) (Lai et al., 2020). DAGMM, DSEBMs, OCGAN and OCSVM were proposed for novelty detection. IF, LOF and RSRAE were originally proposed for outlier detection and we thus apply their trained model for the test data.

For MAW and the above four reconstruction-based methods, that is, DAGMM, DSEBMs, OCGAN and RSRAE, we use the following structure of encoders and decoders, which vary with the type of data (images or non-images). For non-images, which are mapped to feature vectors of dimension $D$, the encoder is a fully connected network with output channels $(32, 64, 128, 128 \times 4)$. The decoder is a fully connected network with output channels $(128, 64, 32, D)$, followed by a normalization layer at the end. For image datasets, the encoder has three convolutional layers with output channels $(32, 64, 128)$, kernel sizes $(5 \times 5, 5 \times 5, 3 \times 3)$ and strides $(2, 2, 2)$. Its output is flattened to lie in $\mathbb{R}^{128}$ and then mapped into a $128 \times 4$ dimensional vector using a dense layer (with output channels $128 \times 4$). The decoder of image datasets first applies a dense layer from $\mathbb{R}^2$ to $\mathbb{R}^{128}$ and then three deconvolutional layers with output channels $(64, 32, 3)$, kernel sizes $(3 \times 3, 5 \times 5, 5 \times 5)$ and strides $(2, 2, 2)$.

For MAW we set the following parameters, where additional details are in §A. Intrinsic dimension: $d = 2$; mixture parameter: $\eta = 5/6$, sampling number: $T = 5$, and size of $\boldsymbol{A}$ (used for dimension reduction): $128 \times 2$. For all experiments, the discriminator is a fully connected network with size $(32, 64, 128, 1)$.

## 4.2 COMPARISON OF MAW WITH STATE-OF-THE-ART METHODS

We use five datasets for novelty detection: KDDCUP-99 (Dua & Graff, 2017), Reuters-21578 (Lewis, 1997), COVID-19 Radiography database (Chowdhury et al., 2020), Caltech101 (Fei-Fei et al., 2004) and Fashion MNIST (Xiao et al., 2017). We distinguish between image datasets (COVID-19, Catlech101 and Fashion MNIST) and non-image datasets (KDDCUP-99 and Reuters-21578). We describe each dataset, common preprocessing procedures and choices of their largest clusters in §F. Each dataset contains several clusters (2 for KDDCUP-99, 5 largest ones for Reuters-21578, 3 for COVID-19, 11 largest ones for Caltech101 and 10 for Fashion MNIST). We arbitrarily fix a class and uniformly sample $N$ training inliers and $N_{\text{test}}$ testing inliers from that class. We let $N = 6000, 350, 160, 100$, $300$ and $N_{\text{test}} = 1200, 140, 60, 100$, $60$ for KDDCUP-99, Reuters-21578, COVID-19, Caltech101 and Fashion MNIST, respectively. We then fix $c$ in $\{0.1, 0.2, 0.3, 0.4, 0.5\}$, and uniformly sample $c$ percentage of outliers from the rest of the clusters for the training data. We also fix $c_{\text{test}}$ in $\{0.1, 0.3, 0.5, 0.7, 0.9\}$ and uniformly sample $c_{\text{test}}$ percentage of outliers from the rest of the clusters for the testing data.

Using all possible thresholds for the finite datasets, we compute the AUC (area under curve) and AP (average precision) scores, while considering the outliers as "positive". For each fixed $c = 0.1, 0.2, 0.3, 0.4, 0.5$ we average these results over the values of $c_{\text{test}}$, the different choices of inlier clusters (among all possible clusters), and three runs with different random initializations for each of these choices. We also compute the corresponding standard deviations. We report these results in Figs. 2 and 3 and further specify numerical values in §H.1. We observe state-of-the-art performance of MAW in all of these datasets. In Reuters-21578, DSEBMs performs slightly better than MAW and OCSVM has comparable performance. However, these two methods are not competitive in the rest of the datasets. In §G, we report results for a different scenario where the outliers of the training and test sets have different characteristics. In this setting, we show that MAW performs even better when compared to other methods.

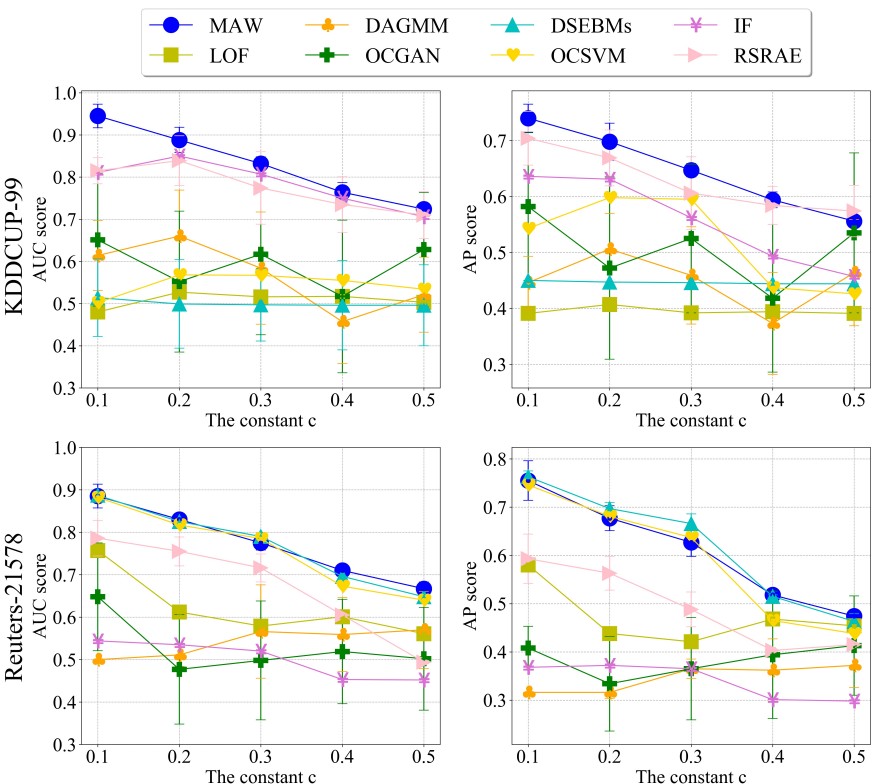

Figure 2: AUC (on left) and AP (on right) scores with training outlier ratios $c = 0.1, 0.2, 0.3, 0.4$ and $0.5$ for the two non-image datasets: KDDCUP-99 and Reuters-21578.

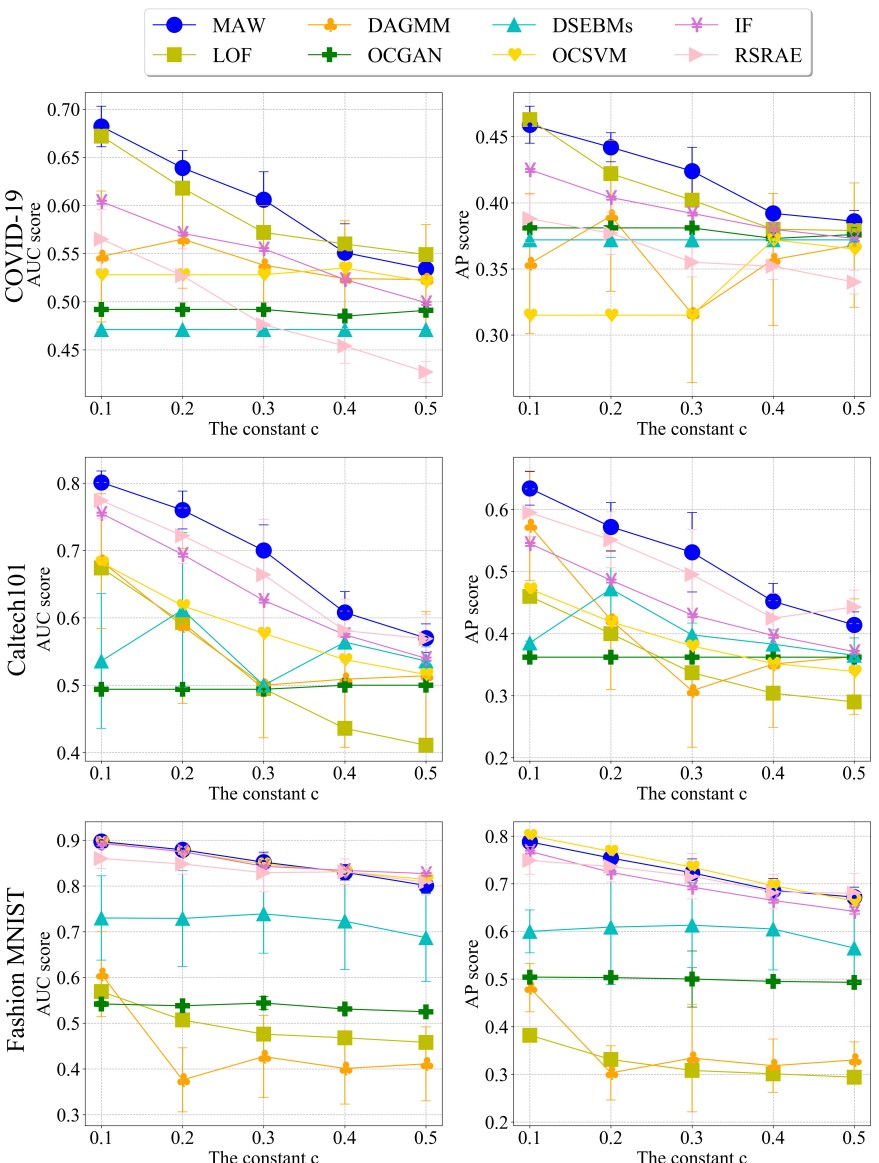

Figure 3: AUC (on left) and AP (on right) scores with training outlier ratios $c = 0.1, 0.2, 0.3, 0.4$ and $0.5$ for the three image datasets: COVID-19, Caltech101 and Fashion MNIST.

### 4.3 TESTING THE EFFECT OF THE NOVEL FEATURES OF MAW

We experimentally validate the effect of the following five new features of MAW: the least absolute deviation for reconstruction, the $W_1$ metric for the regularization of the latent distribution, the Gaussian mixture model assumption, full covariance matrices resulting from dimension reduction component and the lower rank constraint for the inlier mode. The following methods respectively replace each of the above component of MAW with a traditional one: MAW-MSE, MAW-KL divergence, MAW-same rank, MAW-single Gaussian and MAW-diagonal cov., respectively. In addition, we consider a standard variational autoencoder (VAE). Additional details for the latter six methods are in §B.

We compared the above six methods with MAW using two datasets: KDDCUP-99 and COVID-19 with training outlier ratio $c = 0.1, 0.2$ and $0.3$. We followed the experimental setting described in §4.1. Fig. 4 reports the averages and standard deviations of the computed AUC and AP scores, where the corresponding numerical values are further recorded in §H.2. The results indicate a clear decrease of accuracy when missing any of the novel components of MAW or using a standard VAE.

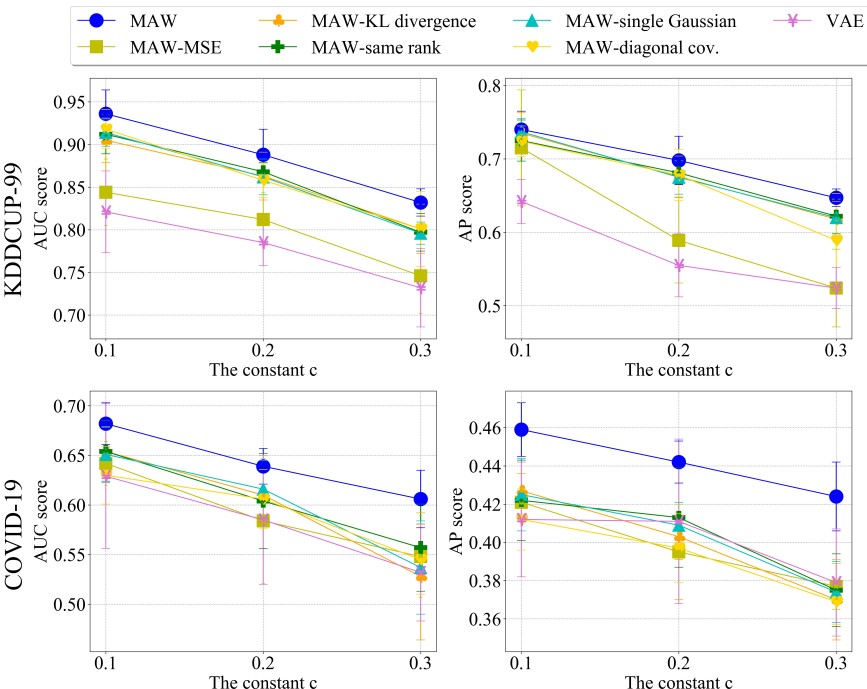

Figure 4: AUC (on left) and AP (on right) scores for variants of MAW (missing a novel component) with training outlier ratios $c = 0.1, 0.2, 0.3$ using the KDDCUP-99 and COVID-19 datasets.

## 5 CONCLUSION AND FUTURE WORK

We introduced MAW, a robust VAE-type framework for novelty detection that can tolerate high corruption of the training data. We proved that the Wasserstein regularization used in MAW has better robustness to outliers and is more suitable to a low-dimensional inlier component than the KL divergence. We demonstrated state-of-the-art performance of MAW with a variety of datasets and experimentally validated that omitting any of the new ideas results in a significant decrease of accuracy.

We hope to further extend our proposal in the following ways. First of all, we plan to extend and test some of our ideas for the different problem of robust generation, in particular, for building generative networks which are robust against adversarial training data. Second of all, we would like to carefully study the virtue of our idea of modeling the most significant mode in a training data. In particular, when extending the work to generation, one has to verify that this idea does not lead to mode collapse. Furthermore, we would like to explore any tradeoff of this idea, as well as our setting of robust novelty detection, with fairness. At last, we hope to further extend our theoretical guarantees. For example, two problems that currently seem intractable are the study of the $W_1$ version of Proposition D.1 and of the minimizer of (14).

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

## A  ADDITIONAL EXPLANATIONS AND IMPLEMENTATION DETAILS OF MAW

In §A.1 we review the ELBO function and explain how ELBOW is obtained from ELBO. Additional implementation details of MAW are in §A.2. At last, §A.3 provides algorithmic boxes for training MAW and applying it for novelty detection.

### A.1  REVIEW OF ELBO AND ITS RELATIONSHIP WITH ELBOW

A standard VAE framework would minimize the expected KL-divergence from $p(\mathbf{z}|\mathbf{x})$ to $q(\mathbf{z}|\mathbf{x})$ in $\mathcal{Q}$, where the expectation is taken over $p(\mathbf{x})$. By Bayes' rule this is equivalent to maximizing the evidence lower bound (ELBO):

$$\text{ELBO}(q) = \mathbb{E}_{p(\mathbf{x})}\mathbb{E}_{q(\mathbf{z}|\mathbf{x})}\log p(\mathbf{x}|\mathbf{z}) - \mathbb{E}_{p(\mathbf{x})}KL(q(\mathbf{z}|\mathbf{x})\|p(\mathbf{z})) .$$

The first term of ELBO is the reconstruction likelihood. Its second term restricts the deviation of $q(\mathbf{z}|\mathbf{x})$ from $p(\mathbf{z})$ and can be viewed as a regularization term. ELBOW is a more robust version of ELBO with a different regularization. That is, it replaces $\mathbb{E}_{p(\mathbf{x})}KL(q(\mathbf{z}|\mathbf{x})\|p(\mathbf{z}))$ with $W_1(q(\mathbf{z}), p(\mathbf{z}))$. We remark that the $W_1$ distance cannot be computed between $q(\mathbf{z}|\mathbf{x})$ and $p(\mathbf{z})$ and ELBOW thus practically replaces $q(\mathbf{z}|\mathbf{x})$ with its expected distribution, $q(\mathbf{z}) = \mathbb{E}_{p(\mathbf{x})}q(\mathbf{z}|\mathbf{x})$ (or a discrete approximation of this).

### A.2  ADDITIONAL IMPLEMENTATION DETAILS OF MAW

The matrix $\boldsymbol{A}$ and the network parameters for encoders, decoders and discriminators are initialized by the Glorot uniform initializer (Glorot & Bengio, 2010).

The neural networks within MAW are implemented with TensorFlow (Abadi et al., 2015) and trained for 100 epochs with batch size 128. We apply batch normalization to each layer of any neural network. The neural networks were optimized by Adam (Kingma & Ba, 2015) with learning rate 0.00005. For the VAE-structure of MAW, we use Adam with learning rate 0.00005. For the WGAN-type structure discriminator of MAW, we perform RMSprop (Bengio & Monperrus, 2005) with learning rate 0.0005, following the recommendation of Arjovsky et al. (2017) for WGAN.

### A.3  ALGORITHMIC BOXES FOR MAW

Algorithms 1 and 2 describe training MAW and applying MAW for novelty detection, respectively. In these descriptions, we denote by $\boldsymbol{\theta}$, $\boldsymbol{\varphi}$ and $\boldsymbol{\delta}$ the trainable parameters of the encoder $\mathcal{E}$, decoder $\mathcal{D}$ and discriminator $\mathcal{D}is$, respectively. Recall that $\boldsymbol{A}$ includes the trained parameters of the dimension reduction component.

---

**Algorithm 1** Training MAW

---

**Input:** Training data $\{\mathbf{x}^{(i)}\}_{i=1}^{L}$; initialized parameters $\boldsymbol{\theta}$, $\boldsymbol{\varphi}$ and $\boldsymbol{\delta}$ of $\mathcal{E}$, $\mathcal{D}$ and $\mathcal{D}is$, respectively; initialized $\boldsymbol{A}$; weight $\eta$; number of epochs; batch size $I$; sampling number $T$; learning rate $\alpha$
**Output:** Trained parameters $\boldsymbol{\theta}$, $\boldsymbol{\varphi}$ and $\boldsymbol{A}$
 1: **for** each epoch **do**
 2:  **for** each batch $\{\mathbf{x}^{(i)}\}_{i\in\mathcal{I}}$ **do**
 3:   $\boldsymbol{\mu}_{0,1}^{(i)}, \boldsymbol{\mu}_{0,2}^{(i)}, \boldsymbol{s}_{0,1}^{(i)}, \boldsymbol{s}_{0,2}^{(i)} \leftarrow \mathcal{E}(\mathbf{x}^{(i)})$
 4:   $\boldsymbol{\mu}_j^{(i)} \leftarrow \boldsymbol{A}^{\mathrm{T}}\boldsymbol{\mu}_{0,j}^{(i)}, \ \boldsymbol{M}_j^{(i)} \leftarrow \boldsymbol{A}^{\mathrm{T}}\mathrm{diag}(\boldsymbol{s}_{0,j}^{(i)})\boldsymbol{A}, \ j = 1, 2$
 5:   Compute $\tilde{\boldsymbol{M}}_1^{(i)}$ according to (5) and (6)
 6:   $\boldsymbol{\Sigma}_1^{(i)} \leftarrow \tilde{\boldsymbol{M}}_1^{(i)}\tilde{\boldsymbol{M}}_1^{(i)\mathrm{T}}, \ \boldsymbol{\Sigma}_2^{(i)} \leftarrow \boldsymbol{M}_2^{(i)}\boldsymbol{M}_2^{(i)\mathrm{T}}$
 7:   **for** $t = 1, \cdots, T$ **do**
 8:     sample a batch $\{\mathbf{z}_{\mathrm{gen}}^{(i,t)}\}_{i\in\mathcal{I}} \sim \eta\mathcal{N}(\boldsymbol{\mu}_1^{(i)}, \boldsymbol{\Sigma}_1^{(i)}) + (1-\eta)\mathcal{N}(\boldsymbol{\mu}_2^{(i)}, \boldsymbol{\Sigma}_2^{(i)})$
 9:     sample a batch $\{\mathbf{z}_{\mathrm{hyp}}^{(i,t)}\}_{i\in\mathcal{I}} \sim \mathcal{N}(\mathbf{0}, \boldsymbol{I})$
10:   **end for**
11:   $(\boldsymbol{\theta}, \boldsymbol{A}, \boldsymbol{\varphi}) \leftarrow (\boldsymbol{\theta}, \boldsymbol{A}, \boldsymbol{\varphi}) - \alpha\nabla_{(\boldsymbol{\theta}, \boldsymbol{A}, \boldsymbol{\varphi})}L_{\mathrm{VAE}}(\boldsymbol{\theta}, \boldsymbol{A}, \boldsymbol{\varphi})$ according to (7)
12:   $\boldsymbol{\delta} \leftarrow \boldsymbol{\delta} - \alpha\nabla_{\boldsymbol{\delta}}L_{W_1}(\boldsymbol{\delta})$ according to (9)
13:   $\boldsymbol{\delta} \leftarrow \mathrm{clip}(\boldsymbol{\delta}, [-1, 1])$
14:   $(\boldsymbol{\theta}, \boldsymbol{A}) \leftarrow (\boldsymbol{\theta}, \boldsymbol{A}) - \alpha\nabla_{(\boldsymbol{\theta}, \boldsymbol{A})}L_{\mathrm{GEN}}(\boldsymbol{\theta}, \boldsymbol{A})$ according to (10)
15:  **end for**
16: **end for**

---

## B MORE DETAILS ON TESTING THE EFFECT OF THE NOVEL FEATURES OF MAW

In §4.3 we experimentally validated the essential components of MAW by implementing variants of MAW that replace each novel component of MAW with a standard one. We notice that the AUC and AP scores in Figs. 2 and 3 consistently decrease when the outlier ratios increase, and thus the chosen training outlier ratios ($c = 0.1, 0.2$ and $0.3$) are sufficient to demonstrate the effectiveness of MAW over its variants. We provide additional details on each of these variants of MAW.

- **MAW-MSE** replaces the least absolute deviation loss $L_{\text{VAE}}$ with the common mean squared error (MSE). That is, it replaces $\left\| \mathbf{x}^{(i)} - \mathcal{D}(\mathbf{z}_{\text{gen}}^{(i,t)}) \right\|_2$ in (7) with $\left\| \mathbf{x}^{(i)} - \mathcal{D}(\mathbf{z}_{\text{gen}}^{(i,t)}) \right\|_2^2$.
- **MAW-KL divergence** replaces the Wasserstein regularization $L_{W_1}$ with the KL-divergence. This is implemented by replacing the WGAN-type structure of the discriminator with a standard GAN.
- **MAW-same rank** uses the same rank $d$ for both the covariance matrices $\boldsymbol{\Sigma}_1^{(i)}$ and $\boldsymbol{\Sigma}_2^{(i)}$, instead of forcing $\boldsymbol{\Sigma}_1^{(i)}$ to have lower rank $d/2$.
- **MAW-single Gaussian** replaces the Gaussian mixture model for the latent distribution with a single Gaussian distribution with a full covariance matrix.
- **MAW-diagonal cov.** replaces the full covariance matrices resulting from the dimension reduction component by diagonal covariances. Its encoder directly produces 2-dimensional means and diagonal covariances (one of rank 1 for the inlier mode and one of rank 2 for the outlier mode).
- **VAE** has the same encoder and decoder structures as MAW. Instead of a dimension reduction component, it uses a dense layer which maps the output of the encoder to a 4-dimensional vector composed of a 2-dimensional mean and 2-dimensional diagonal covariance. This is common for a traditional VAE.

## C SENSITIVITY OF HYPERPARAMETERS

We examine the sensitivity of some of the reported results to changes of some hyperparameters. In §C.1, we report the sensitivity to choices of the intrinsic dimension. In §C.2, we report the sensitivity to choices of the mixture parameter.

---

**Algorithm 2** Applying MAW to novelty detection

---

**Input:** Test data $\{\mathbf{y}^{(j)}\}_{j=1}^N$; sampling number $T$; trained MAW model; threshold $\epsilon_{\text{T}}$; similarity $S(\cdot, \cdot)$
**Output:** Binary labels for novelty for each $j = 1, \ldots, N$

1: **for** $j = 1, \ldots, N$ **do**
2: $\quad \boldsymbol{\mu}_{0,1}^{(j)}, \boldsymbol{s}_{0,1}^{(j)} \leftarrow \mathcal{E}(\mathbf{y}^{(j)})$
3: $\quad \boldsymbol{\mu}_1^{(j)} \leftarrow \boldsymbol{A}^{\text{T}} \boldsymbol{\mu}_{0,1}^{(j)}, \ \boldsymbol{M}_1^{(j)} \leftarrow \boldsymbol{A}^{\text{T}} \text{diag}(\boldsymbol{s}_{0,1}^{(j)}) \boldsymbol{A}$
4: $\quad$ Compute $\tilde{\boldsymbol{M}}_1^{(j)}$ according to (5) and (6)
5: $\quad \boldsymbol{\Sigma}_1^{(j)} \leftarrow \tilde{\boldsymbol{M}}_1^{(j)} \tilde{\boldsymbol{M}}_1^{(j)\text{T}}$
6: $\quad$ **for** $t = 1, \cdots, T$ **do**
7: $\quad\quad$ sample $\mathbf{z}_{\text{in}}^{(j,t)} \sim \mathcal{N}(\boldsymbol{\mu}_1^{(j)}, \boldsymbol{\Sigma}_1^{(j)})$
8: $\quad\quad \tilde{\mathbf{y}}^{(j,t)} \leftarrow \mathcal{D}\left(\mathbf{z}_{\text{in}}^{(j,t)}\right)$
9: $\quad\quad$ compute $S(\mathbf{y}^{(j)}, \tilde{\mathbf{y}}^{(j,t)})$
10: $\quad$ **end for**
11: $\quad S^{(j)} \leftarrow T^{-1} \sum_{t=1}^T S(\mathbf{y}^{(j)}, \tilde{\mathbf{y}}^{(j,t)})$
12: $\quad$ **if** $S^{(j)} \geq \epsilon_{\text{T}}$ **then**
13: $\quad\quad \mathbf{y}^{(j)}$ is a normal example
14: $\quad$ **else**
15: $\quad\quad \mathbf{y}^{(j)}$ is a novelty
16: $\quad$ **end if**
17: **end for**
18: **return** Normality labels for $j = 1, \ldots, N$

---

### C.1 SENSITIVITY TO DIFFERENT INTRINSIC DIMENSIONS

In all of the other experiments in this paper the default value of the intrinsic dimension is $d = 2$. Here we study the sensitivity of our numerical results to the following choices intrinsic dimensions: $d = 2$, $4, 8, 16, 32$ and $64$, while using the KDDCUP-99 and COVID-19 datasets. We fix the intermediate training outlier ratio $c = 0.3$ for demonstration purpose. We compute the AUC and AP scores averaged over testing outlier ratios $c_{\text{test}} = 0.1, 0.3, 0.5, 0.7$ and $0.9$ with three runs per setting. Fig. 5 reports the averaged results and their standard deviations, which are indicated by error bars.

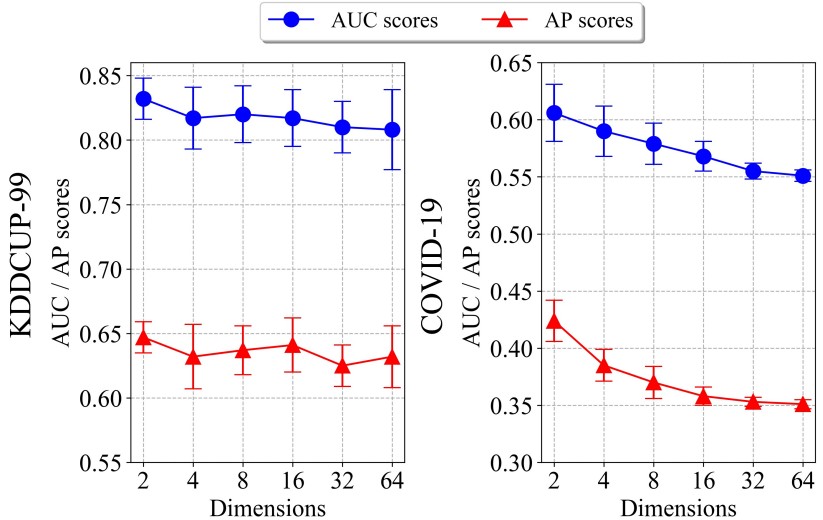

Figure 5: AUC and AP scores with intrinsic dimensions $d = 2, 4, 8, 16, 32$ and $64$ for KDDCUP-99 (on the left) and COVID-19 (on the right), where $c = 0.3$.

We can see from Fig. 5 that our default choice of intrinsic dimension $d = 2$ results in the best performance. For COVID-19 we see a clear decrease of accuracy with the increase of the intrinsic dimension. For KDDCUP-99 we still see a preference for $d = 2$, but the decrease with higher dimensions is not so noticeable as in COVID-19. These experiments confirm our default choice and indicate that the accuracy may decrease when the intrinsic dimension is not sufficiently small.

### C.2 SENSITIVITY TO MIXTURE PARAMETERS

In the rest of our experiments the default value of the mixture parameter $\eta$ is $5/6$. Namely, we assume that the inlier mode has larger weight among the Gaussian mixture. In this section, we study the sensitivity of the accuracy of MAW to the mixture parameters: $\{0.1, 0.2, 0.3, 0.4, 0.5, 0.6, 0.7, 5/6, 0.9\}$. We use $5/6 \approx 0.83$, instead of the nearby value $0.8$, since it was already tested for MAW. The training outlier ratios are $0.1, 0.2$ and $0.3$. We report results on both KDDCUP-99 and COVID-19 in Fig. 6.

We notice that the AUC and AP scores mildly increase as the mixture parameter $\eta$ increases (though they may slightly decrease at $0.9$). It seems that MAW seems to learn the inlier mode better with larger weight for the inlier mode and consequently gain more robustness. Nevertheless, the variation in the accuracy as a function of $\eta$ is not large in general.

## D   ADDITIONAL THEORETICAL GUARANTEES FOR THE $W_1$ MINIMIZATION

In §D.1 we fully motivate our focus on studying (11) in order to understand the advantage of the use of the Wasserstein distance over the KL divergence in the framework of MAW. In §D.2 we prove Proposition 3.1. Additional and more technical proposition that involves low-rank inliers is stated and proved in §D.2.

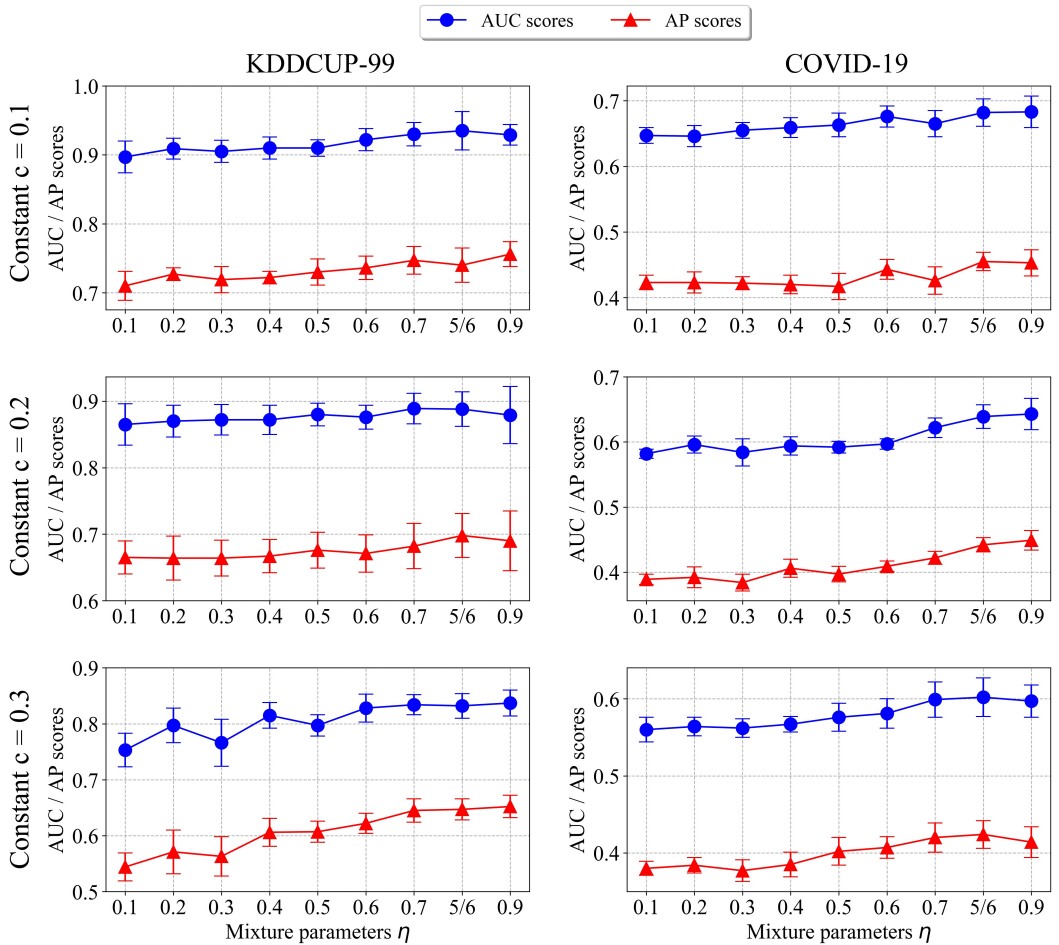

Figure 6: AUC and AP scores with mixture parameters $\eta = 0.1, 0.2, 0.3, 0.4, 0.5, 0.6, 0.7, 5/6$ and $0.9$ for KDDCUP-99 (on the left) and COVID-19 (on the right). From the top to the bottom row, the training outlier ratio are $c = 0.1, 0.2$ and $0.3$, respectively.

### D.1 MOTIVATION OF STUDYING (11)

The implementation of any VAE or its variants, such as AAE, WAE and MAW, requires the optimization of a regularization penalty $\mathcal{R}$, which measures the discrepancy between the latent distribution and the prior distribution. This penalty is typically the KL divergence, though one may use appropriate metrics such as $W_2$ or $W_1$. Therefore, one needs to minimize

$$\mathcal{R}\left(\frac{1}{L}\sum_{i=1}^{L} q(\mathbf{z}|\mathbf{x}^{(i)}), p(\mathbf{z})\right) \tag{12}$$

over the observed variational family $\mathcal{Q} = \{q(\mathbf{z}|\mathbf{x})\}$, which indexed by some parameters of $q$. Here, $L$ is the batch size of the input data and $\sum_{i=1}^{L} q(\mathbf{z}|\mathbf{x}^{(i)})$ is its observed aggregated distribution.

Since the explicit expressions of the regularization measurements between aggregated distributions are unknown, it is not feasible to study the minimizer of (12). We thus consider the following approximation of (12):

$$\sum_{i=1}^{L}\frac{1}{L}\mathcal{R}\left(q(\mathbf{z}|\mathbf{x}^{(i)}), p(\mathbf{z})\right). \tag{13}$$

We can minimize one term of this sum at a time, that, is minimize $\mathcal{R}\left(q(\mathbf{z}|\mathbf{x}), p(\mathbf{z})\right)$ over $\mathcal{Q}$. This minimization strategy is common in the study of the Wasserstein barycenter problem (Agueh & Carlier, 2011; Peyré et al., 2019; Chen et al., 2018).

One of the underlying assumptions of MAW is that the prior distribution $p(\mathbf{z})$ is Gaussian and $q(\mathbf{z}|\mathbf{x})$ is a Gaussian mixture. That is, $p(\mathbf{z}) \sim \mathcal{N}(\boldsymbol{\mu_0}, \boldsymbol{\Sigma}_0)$ and $q(\mathbf{z}|\mathbf{x}) \sim \eta\mathcal{N}(\boldsymbol{\mu}_1, \boldsymbol{\Sigma}_1) + (1-\eta)\mathcal{N}(\boldsymbol{\mu}_2, \boldsymbol{\Sigma}_2)$. This gives rise to the following minimization problem

$$\min_{\boldsymbol{\mu}_1, \boldsymbol{\mu}_2 \in \mathbb{R}^K; \boldsymbol{\Sigma}_1, \boldsymbol{\Sigma}_2 \in \mathcal{S}_+^K} \mathcal{R}\left(\eta\mathcal{N}(\boldsymbol{\mu}_1, \boldsymbol{\Sigma}_1) + (1-\eta)\mathcal{N}(\boldsymbol{\mu}_2, \boldsymbol{\Sigma}_2), \mathcal{N}(\boldsymbol{\mu}_0, \boldsymbol{\Sigma}_0)\right). \tag{14}$$

Similarly to approximating (12) by (13), we approximate (14) by the following minimization problem:

$$\min_{\boldsymbol{\mu}_1, \boldsymbol{\mu}_2 \in \mathbb{R}^K; \boldsymbol{\Sigma}_1, \boldsymbol{\Sigma}_2 \in \mathcal{S}_+^K} \eta\mathcal{R}\left(\mathcal{N}(\boldsymbol{\mu}_1, \boldsymbol{\Sigma}_1), \mathcal{N}(\boldsymbol{\mu}_0, \boldsymbol{\Sigma}_0)\right) + (1-\eta)\mathcal{R}\left(\mathcal{N}(\boldsymbol{\mu}_2, \boldsymbol{\Sigma}_2), \mathcal{N}(\boldsymbol{\mu}_0, \boldsymbol{\Sigma}_0)\right).$$

Recall that in MAW $\mathcal{N}(\boldsymbol{\mu}_1, \boldsymbol{\Sigma}_1)$ and $\mathcal{N}(\boldsymbol{\mu}_2, \boldsymbol{\Sigma}_2)$ are associated with the inlier and outlier distribution of MAW. We further assume that there is a sufficiently small threshold $\epsilon > 0$ for which $\|\boldsymbol{\mu}_1 - \boldsymbol{\mu}_2\|_2 \geq \epsilon$. This is a reasonable assumption since, in practice, if $\boldsymbol{\mu}_1$ and $\boldsymbol{\mu}_2$ are very close, the reconstruction loss will be large. These assumptions lead to the optimization problem (11) proposed in §3.

## D.2   PROOF OF PROPOSITION 3.1

Recall that $\boldsymbol{\mu}_0 \in \mathbb{R}^K$ is the mean of the prior Gaussian, $\epsilon > 0$ is the fixed separation parameter for the means of the two modes and $\eta > 1/2$ is the fixed mixture parameter. For $i = 0, 1, 2$, we denote the Gaussian probability distribution by $\mathcal{N}(\boldsymbol{\mu}_i, \boldsymbol{\Sigma}_i)$. Since in our setting $\boldsymbol{\Sigma}_0 = \boldsymbol{\Sigma}_1 = \boldsymbol{\Sigma}_2$, we denote the common covariance matrix in $\mathcal{S}_{++}^K$ by $\boldsymbol{\Sigma}$. That is, $\boldsymbol{\Sigma} = \boldsymbol{\Sigma}_i$ for $i = 0, 1, 2$.

We first analyze the solution of (11) with $\mathcal{R} = W_p$, where $p \geq 1$, and then analyze the solution of (11) with $\mathcal{R} = KL$.

**The case $\mathcal{R} = W_p, p \geq 1$:** We follow the next three steps to prove that the minimizer of (11) satisfies $\boldsymbol{\mu}_1 = \boldsymbol{\mu}_0$.

**Step I:** We prove that

$$W_p(\nu_i, \nu_0) \equiv W_p(\mathcal{N}(\boldsymbol{\mu}_i, \boldsymbol{\Sigma}), \mathcal{N}(\boldsymbol{\mu}_0, \boldsymbol{\Sigma})) = \|\boldsymbol{\mu}_i - \boldsymbol{\mu}_0\|_2 \text{ for } p \geq 1 \text{ and } i = 1, 2. \tag{15}$$

First, we note that using the definition of $W_p, p \geq 1$ and the common notation $\Pi(\nu_i, \nu_0)$ for the distribution on $\mathbb{R}^K \times \mathbb{R}^K$ with marginals $\nu_i$ and $\nu_0$

$$\begin{aligned} W_p^p(\nu_i, \nu_0) &= \inf_{\pi \in \Pi(\nu_i, \nu_0)} \mathbb{E}_{(\mathbf{x}, \mathbf{y}) \sim \pi} \|\mathbf{x} - \mathbf{y}\|_2^p \\ &\geq \inf_{\pi \in \Pi(\nu_i, \nu_0)} \left\|\mathbb{E}_{(\mathbf{x}, \mathbf{y}) \sim \pi}\mathbf{x} - \mathbb{E}_{(\mathbf{x}, \mathbf{y}) \sim \pi}\mathbf{y}\right\|_2^p \\ &= \|\boldsymbol{\mu}_i - \boldsymbol{\mu}_0\|_2^p, \end{aligned} \tag{16}$$

where the inequality follows the fact that $\|.\|_2^p$ is convex and from Jensen's inequality.

On the other hand, for $i = 1$ or $i = 2$, let $\mathbf{x}^*$ be an arbitrary random vector with distribution $\nu_i$, and let $\mathbf{y}^* = \mathbf{x}^* - \boldsymbol{\mu}_i + \boldsymbol{\mu}_0$. The distribution of $\mathbf{y}^*$ is Gaussian with mean $\boldsymbol{\mu}_0$ and covariance $\boldsymbol{\Sigma}_i$, that is, this distribution is $\nu_0$. Let $\pi^*$ be the joint distribution of the random variables $\mathbf{x}^*$ and $\mathbf{y}^*$. We note that $\pi^*$ is in $\Pi(\nu_i, \nu_0)$ and that

$$\mathbb{E}_{(\mathbf{x}, \mathbf{y}) \sim \pi^*} \|\mathbf{x} - \mathbf{y}\|_2^p = \mathbb{E}_{(\mathbf{x}, \mathbf{y}) \sim \pi^*} \|\boldsymbol{\mu}_i - \boldsymbol{\mu}_0\|_2^p = \|\boldsymbol{\mu}_i - \boldsymbol{\mu}_0\|_2^p.$$

Therefore,

$$W_p^p(\nu_i, \nu_0) = \inf_{\pi \in \Pi(\nu_i, \nu_0)} \mathbb{E}_{(\mathbf{x}, \mathbf{y}) \sim \pi} \|\mathbf{x} - \mathbf{y}\|_2^p \leq \mathbb{E}_{(\mathbf{x}, \mathbf{y}) \sim \pi^*} \|\mathbf{x} - \mathbf{y}\|_2^p = \|\boldsymbol{\mu}_i - \boldsymbol{\mu}_0\|_2^p. \tag{17}$$

The combination of (16) and (17) immediately yields (15).

**Step II:** We prove that (11) with $\mathcal{R} = W_p, p \geq 1$, is equivalent to

$$\min_{\substack{\boldsymbol{\mu}_1, \boldsymbol{\mu}_2 \in \mathbb{R}^K; \\ \text{s.t. } \boldsymbol{\mu}_0, \boldsymbol{\mu}_1, \boldsymbol{\mu}_2: \text{colinear} \\ \& \|\boldsymbol{\mu}_1 - \boldsymbol{\mu}_2\|_2 \geq \epsilon}} \eta \|\boldsymbol{\mu}_1 - \boldsymbol{\mu}_0\|_2 + (1-\eta) \|\boldsymbol{\mu}_2 - \boldsymbol{\mu}_0\|_2. \tag{18}$$

We first note that (11) with $\mathcal{R} = W_p, p \geq 1$ is equivalent to

$$\min_{\substack{\boldsymbol{\mu}_1, \boldsymbol{\mu}_2 \in \mathbb{R}^K \\ \text{s.t. } \|\boldsymbol{\mu}_1 - \boldsymbol{\mu}_2\|_2 \geq \epsilon}} \eta \|\boldsymbol{\mu}_1 - \boldsymbol{\mu}_0\|_2 + (1-\eta) \|\boldsymbol{\mu}_2 - \boldsymbol{\mu}_0\|_2. \tag{19}$$

Indeed, this is a direct consequence of the expression derived in step I for $\mathcal{R}$ in this case. It is thus left to show that if $\boldsymbol{\mu}_1', \boldsymbol{\mu}_2' \in \mathbb{R}^K$ minimize (19), then we can construct $\widetilde{\boldsymbol{\mu}_1'}, \widetilde{\boldsymbol{\mu}_2'} \in \mathbb{R}^K$ that are colinear with $\boldsymbol{\mu}_0$ and also minimize (19).

For any $\boldsymbol{\mu_1}$ and $\boldsymbol{\mu_2}$ in $\mathbb{R}^K$ with $\|\boldsymbol{\mu}_1 - \boldsymbol{\mu}_2\|_2 \geq \epsilon$ and for the given $\boldsymbol{\mu}_0 \in \mathbb{R}^K$, we define $\tilde{\boldsymbol{\mu}}_0, \tilde{\boldsymbol{\mu}}_1$ and $\tilde{\boldsymbol{\mu}}_2 \in \mathbb{R}^K$ and demonstrate them in Fig. 7. The point $\tilde{\boldsymbol{\mu}}_0$ is the projection of $\boldsymbol{\mu}_0$ onto $\boldsymbol{\mu}_1 - \boldsymbol{\mu}_2$ and

$\tilde{\boldsymbol{\mu}}_i := \boldsymbol{\mu}_i + \boldsymbol{\mu}_0 - \tilde{\boldsymbol{\mu}}_0$ for $i = 1, 2$. We observe the following properties, which can be proved by direct calculation, though Fig. 7 also clarifies them:

$$\|\boldsymbol{\mu}_i - \boldsymbol{\mu}_0\|_2 \geq \|\tilde{\boldsymbol{\mu}}_i - \boldsymbol{\mu}_0\|_2 \text{ for } i = 1, 2,$$

and consequently,

$$\eta \|\boldsymbol{\mu}_1 - \boldsymbol{\mu}_0\|_2 + (1 - \eta) \|\boldsymbol{\mu}_2 - \boldsymbol{\mu}_0\|_2 \geq \eta \|\tilde{\boldsymbol{\mu}}_1 - \boldsymbol{\mu}_0\|_2 + (1 - \eta) \|\tilde{\boldsymbol{\mu}}_2 - \boldsymbol{\mu}_0\|_2 ; \tag{20}$$

$$\|\tilde{\boldsymbol{\mu}}_1 - \tilde{\boldsymbol{\mu}}_2\|_2 = \|\boldsymbol{\mu}_1 - \boldsymbol{\mu}_2\|_2 \geq \epsilon; \tag{21}$$

and

$$\tilde{\boldsymbol{\mu}}_1, \ \tilde{\boldsymbol{\mu}}_2, \ \text{and } \boldsymbol{\mu}_0 \text{ are colinear.} \tag{22}$$

Clearly, the combination of (20), (21) and (22) concludes the proof of step II. That is, it implies that if $\boldsymbol{\mu}'_1$, $\boldsymbol{\mu}'_2 \in \mathbb{R}^K$ minimize (19), then $\widetilde{\boldsymbol{\mu}'_1}$ and $\widetilde{\boldsymbol{\mu}'_2}$ defined above are colinear with $\boldsymbol{\mu}_0$ and also minimize (19).

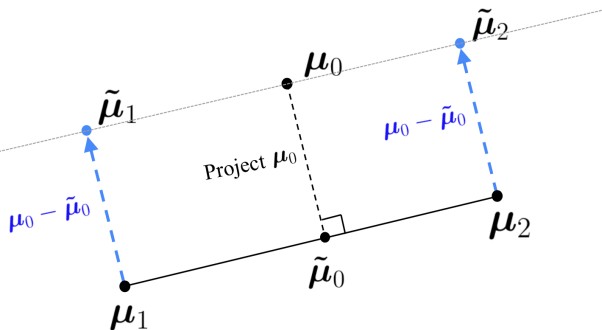

Figure 7: Illustration of the points $\tilde{\boldsymbol{\mu}}_0$, $\tilde{\boldsymbol{\mu}}_1$ and $\tilde{\boldsymbol{\mu}}_2$ and their properties.

**Step III:** We directly solve (18) and consequently (11) with $\mathcal{R} = W_p, p \geq 1$. Due to the colinearity constraint in (11), we can write

$$\boldsymbol{\mu_0} = (1 + t)\boldsymbol{\mu_1} - t\boldsymbol{\mu_2} \text{ for } t \in \mathbb{R}. \tag{23}$$

The objective function in (18) can then be written as

$$\|\boldsymbol{\mu}_1 - \boldsymbol{\mu}_2\|_2 \left( \eta |t| + (1 - \eta)|1 + t| \right) \geq \epsilon \left( \eta |t| + (1 - \eta)|1 + t| \right),$$

where equality is achieved if and only if $\|\boldsymbol{\mu}_1 - \boldsymbol{\mu}_2\|_2 = \epsilon$. We thus define $r(t) = \eta |t| + (1 - \eta)|1 + t|$ and note that

$$r(t) = \begin{cases} t + (1 - \eta), & t \geq 0 \\ (1 - 2\eta)t + (1 - \eta), & 0 \geq t \geq -1 \\ -t + (\eta - 1), & -1 \geq t \end{cases}$$

and its derivative is

$$r'(t) = \begin{cases} 1, & t > 0 \\ 1 - 2\eta, & 0 > t > -1 \\ -1, & -1 > t. \end{cases}$$

The above expressions for $r$ and $r'$ and the assumption that $\eta > 1/2$ imply that $r(t)$ is increasing when $t > 0$, decreasing when $t < 0$ and $r(0) = 1 - \eta < \eta = r(1)$. Thus $r$ has a global minimum at $t = 0$. Hence, it follows from (23) that the minimizer of (11), and equivalently (11) with $\mathcal{R} = W_p, p \geq 1$ satisfies $\boldsymbol{\mu}_1 = \boldsymbol{\mu}_0$.

**The case $\mathcal{R} = KL$:** We prove that the solution of (11) with $\mathcal{R} = KL$ satisfies $\boldsymbol{\mu}_0 = \eta\boldsymbol{\mu}_1 + (1 - \eta)\boldsymbol{\mu}_2$. We practically follow similar steps as the proof above.

**Step I:** We derive an expression for $KL(\nu_i||\nu_0)$, where $i = 1, 2$. We use the following general formula, which holds for the case where $\boldsymbol{\Sigma}_0, \boldsymbol{\Sigma}_1$ and $\boldsymbol{\Sigma}_2$ are general covariance matrices in $\mathcal{S}^K_{++}$ (see e.g., (2) in Hershey & Olsen (2007)):

$$KL(\nu_i||\nu_0) = \frac{1}{2} \left( \log \frac{\det \boldsymbol{\Sigma}_0}{\det \boldsymbol{\Sigma}_i} - K + \text{tr}(\boldsymbol{\Sigma}_0^{-1}\boldsymbol{\Sigma}_i) + (\boldsymbol{\mu}_i - \boldsymbol{\mu}_0)^{\text{T}}\boldsymbol{\Sigma}_0^{-1}(\boldsymbol{\mu}_i - \boldsymbol{\mu}_0) \right) . \tag{24}$$

Since in our setting $\boldsymbol{\Sigma}_1 = \boldsymbol{\Sigma}_2 = \boldsymbol{\Sigma}$, this expression has the simpler form:

$$KL(\nu_i||\nu_0) = \frac{1}{2}(\boldsymbol{\mu}_i - \boldsymbol{\mu}_0)^{\text{T}}\boldsymbol{\Sigma}^{-1}(\boldsymbol{\mu}_i - \boldsymbol{\mu}_0).$$

**Step II:** We reformulate the optimization problem. The above step imples that (11) with $\mathcal{R} = KL$ can be written as

$$\min_{\|\boldsymbol{\mu}_1 - \boldsymbol{\mu}_2\|_2 \geq \epsilon} \eta(\boldsymbol{\mu}_1 - \boldsymbol{\mu}_0)^{\mathrm{T}} \boldsymbol{\Sigma}^{-1} (\boldsymbol{\mu}_1 - \boldsymbol{\mu}_0) + (1 - \eta)(\boldsymbol{\mu}_2 - \boldsymbol{\mu}_0)^{\mathrm{T}} \boldsymbol{\Sigma}^{-1} (\boldsymbol{\mu}_2 - \boldsymbol{\mu}_0),$$

or equivalently,

$$\min_{\|\boldsymbol{\mu}_1 - \boldsymbol{\mu}_2\|_2 \geq \epsilon} \eta \left\| \boldsymbol{\Sigma}^{-\frac{1}{2}} (\boldsymbol{\mu}_1 - \boldsymbol{\mu}_0) \right\|_2^2 + (1 - \eta) \left\| \boldsymbol{\Sigma}^{-\frac{1}{2}} (\boldsymbol{\mu}_2 - \boldsymbol{\mu}_0) \right\|_2^2. \tag{25}$$

We express the eigenvalue decomposition of $\boldsymbol{\Sigma}^{-1}$ as $\boldsymbol{\Sigma}^{-1} = \boldsymbol{U} \boldsymbol{\Lambda} \boldsymbol{U}^{\mathrm{T}}$, where $\boldsymbol{\Lambda} \in \mathcal{S}_+^K$, and $\boldsymbol{U}$ is an orthogonal matrix. Applying the change of variables $\boldsymbol{\mu}_i' = \boldsymbol{\Lambda}^{\frac{1}{2}} \boldsymbol{U}^{\mathrm{T}} \boldsymbol{\mu}_i$ for $i = 0, 1, 2$, we rewrite (25) as

$$\min_{\left\| \boldsymbol{\mu}_1' - \boldsymbol{\mu}_2' \right\|_2 \geq \epsilon} \eta \left\| \boldsymbol{\mu}_1' - \boldsymbol{\mu}_0' \right\|_2^2 + (1 - \eta) \left\| \boldsymbol{\mu}_2' - \boldsymbol{\mu}_0' \right\|_2^2. \tag{26}$$

At last, applying the same colinearity argument as above (supported by Fig. 7) we conclude the following equivalent formulation of (26):

$$\min_{\substack{\boldsymbol{\mu}_0', \boldsymbol{\mu}_1', \boldsymbol{\mu}_2' \text{ are colinear} \\ \& \left\| \boldsymbol{\mu}_1' - \boldsymbol{\mu}_2' \right\|_2 \geq \epsilon}} \eta \left\| \boldsymbol{\mu}_1' - \boldsymbol{\mu}_0' \right\|_2^2 + (1 - \eta) \left\| \boldsymbol{\mu}_2' - \boldsymbol{\mu}_0' \right\|_2^2 \tag{27}$$

**Step III:** We directly solve (27). Due to the colinearity constraint, we can write

$$\boldsymbol{\mu}_0' = (1 + t) \boldsymbol{\mu}_1' - t \boldsymbol{\mu}_2' \text{ for } t \in \mathbb{R} \tag{28}$$

and express the objective function of (27) as

$$\left\| \boldsymbol{\mu}_1' - \boldsymbol{\mu}_2' \right\|_2^2 \left( \eta t^2 + (1 - \eta)(1 + t)^2 \right) \geq \epsilon^2 \left( \eta t^2 + (1 - \eta)(1 + t)^2 \right),$$

where equality is achieved if and only if $\| \boldsymbol{\mu}_1' - \boldsymbol{\mu}_2' \|_2 = \epsilon$. We thus define $r(t) = \eta t^2 + (1 - \eta)(1 + t)^2$ and note that $r'(t) = 2(t + (1 - \eta))$ and $r''(t) = 2$, and thus conclude that $r(t)$ obtains its global minimum at $t = \eta - 1$. This observation and (28) imply that the minimizers $\boldsymbol{\mu}_1$ and $\boldsymbol{\mu}_2$ of (11) with $\mathcal{R} = KL$ satisfy $\boldsymbol{\mu}_0 = \eta \boldsymbol{\mu}_1 + (1 - \eta) \boldsymbol{\mu}_2$.

### D.3 THEORETICAL GUARANTEES FOR (11) WITH LOW-RANK $\boldsymbol{\Sigma}_1$

We study the minimization problem (11) when $\boldsymbol{\Sigma}_1$ has low rank and $\boldsymbol{\Sigma}_2 \in \mathcal{S}_{++}^K$, and also when $\mathcal{R} = W_2$ or $\mathcal{R} = KL$. Unfortunately, the case where $\mathcal{R} = W_1$ is hard to analyze and compute. We first formulate our result for $\mathcal{R} = W_2$. In this case we assume that the prior distribution is a standard Gaussian distribution on $\mathbb{R}^K$. That is, it has mean $\boldsymbol{\mu}_0 = \boldsymbol{0}_K$ and covariance $\boldsymbol{\Sigma}_0 = \boldsymbol{I}_{K \times K}$. We further denote by $\boldsymbol{1}_K$ the vector $(1, \cdots, 1) \in \mathbb{R}^K$. Similarly, we may define for any $n \in \mathbb{N}$, $\boldsymbol{0}_n, \boldsymbol{1}_n, \boldsymbol{I}_{n \times n}$. When it is clear from the context we only use $\boldsymbol{0}, \boldsymbol{1}$ and $\boldsymbol{I}$. For vectors $\boldsymbol{a} \in \mathbb{R}^n$ and $\boldsymbol{b} \in \mathbb{R}^m$, we denote the concatenated vector in $\mathbb{R}^{n+m}$ by $(\boldsymbol{a}; \boldsymbol{b})$.

**Proposition D.1.** *If $\kappa, K \in \mathbb{N}, K > \kappa \geq 1, \epsilon > 0, 1 > \eta > \eta^\star := \frac{K - \kappa + \epsilon^2}{K - \kappa + 2\epsilon^2}, u^\star := \left( \frac{(K - \kappa)(1 - \eta)}{\epsilon^2 (2\eta - 1)} \right)^{\frac{1}{3}}$, where one can note that $\eta^\star > \frac{1}{2}$ and $u^\star \in (0, 1)$), then the minimizer of (11) with $\mathcal{R} = W_2$ and with the constraint that $\boldsymbol{\Sigma}_1$ is of rank $\kappa$ and $\boldsymbol{\Sigma}_2$ is of rank $K$, or equivalently, the minimizer of*

$$\min_{\substack{\boldsymbol{\mu}_1, \boldsymbol{\mu}_2 \in \mathbb{R}^K; \boldsymbol{\Sigma}_1 \in \mathcal{S}_+^K, \boldsymbol{\Sigma}_2 \in \mathcal{S}_{++}^K \\ \text{s.t. } \|\boldsymbol{\mu}_1 - \boldsymbol{\mu}_2\|_2 \geq \epsilon \\ \& \text{ rank}(\boldsymbol{\Sigma}_1) = \kappa, \text{ rank}(\boldsymbol{\Sigma}_2) = K}} \eta W_2(\mathcal{N}(\boldsymbol{\mu}_1, \boldsymbol{\Sigma}_1), \mathcal{N}(\boldsymbol{0}, \boldsymbol{I})) + (1 - \eta) W_2(\mathcal{N}(\boldsymbol{\mu}_2, \boldsymbol{\Sigma}_2), \mathcal{N}(\boldsymbol{0}, \boldsymbol{I})) \tag{29}$$

*satisfies $\boldsymbol{0}_K = u^\star \boldsymbol{\mu}_2 + (1 - u^\star) \boldsymbol{\mu}_1$, $\boldsymbol{\Sigma}_1 = \mathrm{diag}(\boldsymbol{1}_\kappa; \boldsymbol{0}_{K-\kappa})$ and $\boldsymbol{\Sigma}_2 = \mathrm{diag}(\boldsymbol{1}_\kappa; (u^\star)^{-2} \boldsymbol{1}_{K-\kappa})$. Moreover, $\|\boldsymbol{\mu}_1\|_2 = u^\star \epsilon$ and $\|\boldsymbol{\mu}_2\|_2 = (1 - u^\star) \epsilon$.*

We next formulate our simple result on the ill-posedness of (11) with $\mathcal{R} = W_2$ and with the same constraint as in Proposition D.1.

**Proposition D.2.** *If $\kappa, K \in \mathbb{N}, K > \kappa \geq 1, \epsilon > 0, \eta > 0, \boldsymbol{\mu}_0, \boldsymbol{\mu}_1 \in \mathbb{R}^K, \boldsymbol{\Sigma}_0 \in \mathcal{S}_{++}^K$ and $\boldsymbol{\Sigma}_1 \in \mathcal{S}_+^K$ with $\mathrm{rank}(\boldsymbol{\Sigma}) = \kappa$, then*

$$KL(\mathcal{N}(\boldsymbol{\mu}_1, \boldsymbol{\Sigma}_1) \| \mathcal{N}(\boldsymbol{\mu}_0, \boldsymbol{\Sigma}_0)) = \infty.$$

*Therefore, the solution of (11) with $\mathcal{R} = KL$ with the additional constraint that $\boldsymbol{\Sigma}_1$ is of rank $\kappa$ and $\boldsymbol{\Sigma}_0 = \boldsymbol{I}$ is ill-posed.*

Next we clarify the implications of both propositions. Note that Proposition D.1 implies that as $\eta \to 1, u^\star \to 0$. Hence for the inlier component $\boldsymbol{\mu}_1 \to \boldsymbol{0}_K$ as $\eta \to 1$ and $\boldsymbol{\Sigma}_1 = \mathrm{diag}(\boldsymbol{1}_\kappa; \boldsymbol{0}_{K-\kappa})$, so in the limit the inlier distribution has the same mean as the prior distribution and, independently of

$\eta$, its covariance is obtained by an appropriate projection of the covariance $\mathbf{\Sigma}_0$ onto a $\kappa$-dimensional subspace. We similarly note that as $\eta \to 1$, $\mathbf{\Sigma}_2 \to \mathrm{diag}(\mathbf{1}_\kappa; \boldsymbol{\infty}_{K-k})$, so that the outliers will disperse.

We further note that Proposition D.2 implies that the KL divergence fails is unsuitable for low-rank covariance modeling as it leads to an infinite value in the optimization problem.

At last, we note that the inlier and outlier covariances, $\mathbf{\Sigma}_1$ and $\mathbf{\Sigma}_2$, obtained by Proposition D.1, are diagonal. Furthermore, the proof of Proposition D.1 clarifies that the underlying minimization problem of this proposition may assume without loss of generality that the inlier and outlier covariances are diagonal (see e.g., (32), which is formulated below). On the other hand, the numerical results in §4.3 support the use of full covariances, instead of diagonal covariance. Nonetheless, we claim that the full covariances matrices of MAW comes naturally from the dimension reduction component of MAW. This component also contains trainable parameters for the covariances and they will effect the weights of the encoder, that is, will effect both the $W_1$ minimization and the reconstruction loss. Thus the analysis of the $W_1$ minimization component is not sufficient for inferring the whole behavior of MAW. For tractability purposes, the minimization in (11) ignores the dimension reduction component. For completeness we remark that there are two other differences between the use of (11) in Proposition D.1 and the way it arises in MAW that may possibly also result in the advantage of using full covariance in MAW. First of all, the minimization in Proposition D.1 uses $\mathcal{R} = W_2$, whereas MAW uses $\mathcal{R} = W_1$, which we find intractable when using the rest of the setting of Proposition D.1. Second of all, the optimization problem (11) with $\mathcal{R} = W_1$ is an approximation of the minimization of $W_1\left(\frac{1}{L}\sum_{i=1}^{L} q(\mathbf{z}|\mathbf{x}^{(i)}), p(\mathbf{z})\right)$ (see §D.1 for explanation), which is also intractable (even if one uses $\mathcal{R} = W_2$).

In §D.4 we prove Proposition D.1 and in §D.5 we prove Proposition D.2.

### D.4 PROOF OF PROPOSITION D.1

We follow the same steps of the proof of Proposition 3.1

**Step I:** We immediately verify the formula

$$W_2(\mathcal{N}(\boldsymbol{\mu}_i, \mathbf{\Sigma}_i), \mathcal{N}(\mathbf{0}, \boldsymbol{I})) = \sqrt{\|\boldsymbol{\mu}_i\|_2^2 + \left\|\mathbf{\Sigma}_i^{\frac{1}{2}} - \boldsymbol{I}\right\|_F^2} \text{ for } i = 1, 2. \tag{30}$$

We use the following general formula, which holds for the case where $\mathbf{\Sigma}_0$, $\mathbf{\Sigma}_1$ and $\mathbf{\Sigma}_2$ are general covariance matrices in $\mathcal{S}_+^K$ (see e.g., (4) in Panaretos & Zemel (2019)):

$$W_2^2(\mathcal{N}(\boldsymbol{\mu}_i, \mathbf{\Sigma}_i), \mathcal{N}(\boldsymbol{\mu}_0, \mathbf{\Sigma}_0)) = \|\boldsymbol{\mu}_i - \boldsymbol{\mu}_0\|_2^2 + \mathrm{tr}(\mathbf{\Sigma}_i + \mathbf{\Sigma}_0 - 2(\mathbf{\Sigma}_i^{\frac{1}{2}}\mathbf{\Sigma}_0\mathbf{\Sigma}_i^{\frac{1}{2}})^{\frac{1}{2}}), \ i = 1, 2. \tag{31}$$

Indeed, (30) is obtained as a direct consequence of (31) using the identity

$$\mathrm{tr}\left(\mathbf{\Sigma}_i + \boldsymbol{I} - 2\mathbf{\Sigma}_i^{\frac{1}{2}}\right) = \mathrm{tr}\left(\left(\mathbf{\Sigma}_i^{\frac{1}{2}} - \boldsymbol{I}\right)^2\right) = \left\|\mathbf{\Sigma}_i^{\frac{1}{2}} - \boldsymbol{I}\right\|_F^2.$$

**Step II:** We reformulate the underlying minimization problem in two different stages. We first claim that the minimizer of (11) with $\mathcal{R} = W_2$ and the constraint that $\mathbf{\Sigma}_1$ is of rank $\kappa$ and $\mathbf{\Sigma}_2$ is of rank $K$ can be expressed as the minimizer of

$$\min_{\substack{\boldsymbol{\mu}_1, \boldsymbol{\mu}_2 \in \mathbb{R}^K \text{ s.t. } \|\boldsymbol{\mu}_1 - \boldsymbol{\mu}_2\|_2 = \epsilon, \\ \mathbf{\Sigma}_1, \mathbf{\Sigma}_2 \text{ diagonal in } \mathbb{R}^{K \times K} \\ \& \text{ rank}(\mathbf{\Sigma}_1) = \kappa, \text{ rank}(\mathbf{\Sigma}_2) = K}} \eta\sqrt{\|\boldsymbol{\mu}_1\|_2^2 + \left\|\mathbf{\Sigma}_1^{\frac{1}{2}} - \boldsymbol{I}\right\|_F^2} + (1-\eta)\sqrt{\|\boldsymbol{\mu}_2\|_2^2 + \left\|\mathbf{\Sigma}_2^{\frac{1}{2}} - \boldsymbol{I}\right\|_F^2}. \tag{32}$$

In view of (11) and (30) we only need to prove that the minimizer of (32) is the same if one removes the constraint that $\mathbf{\Sigma}_1$ and $\mathbf{\Sigma}_2$ are both diagonal matrices and require instead that they are in $\in \mathcal{S}_+^K$. This is easy to show. Indeed, if for $i = 1$ or $i = 2$, $\mathbf{\Sigma}_i \in \mathcal{S}_+^K$, then it can be diagonalized as follows: $\mathbf{\Sigma}_i = \boldsymbol{U}_i^{\mathrm{T}}\boldsymbol{\Lambda}_i\boldsymbol{U}_i$, where $\boldsymbol{\Lambda}_i \in \mathcal{S}_+^K$ is diagonal and $\boldsymbol{U}_i$ is orthogonal. Hence, $\mathbf{\Sigma}_i^{\frac{1}{2}} = \boldsymbol{U}_i^{\mathrm{T}}\boldsymbol{\Lambda}_i^{\frac{1}{2}}\boldsymbol{U}_i$ and $\left\|\mathbf{\Sigma}_i^{\frac{1}{2}} - \boldsymbol{I}\right\|_F^2 = \left\|\boldsymbol{U}_i^{\mathrm{T}}\boldsymbol{\Lambda}_i^{\frac{1}{2}}\boldsymbol{U}_i - \boldsymbol{I}\right\|_F^2 = \left\|\boldsymbol{U}_i^{\mathrm{T}}(\boldsymbol{\Lambda}_i^{\frac{1}{2}} - \boldsymbol{I})\boldsymbol{U}_i\right\|_F^2 = \left\|\boldsymbol{\Lambda}_i^{\frac{1}{2}} - \boldsymbol{I}\right\|_F^2$. Consequently,

$$W_2(\mathcal{N}(\boldsymbol{\mu}_i, \mathbf{\Sigma}_i), \mathcal{N}(\mathbf{0}, \boldsymbol{I})) = W_2(\mathcal{N}(\boldsymbol{\mu}_i, \boldsymbol{\Lambda}_i), \mathcal{N}(\mathbf{0}, \boldsymbol{I})) \text{ for } i = 1, 2,$$

and the above claim is concluded.

Next, we vectorize the minimization problem in (32) as follows. We denote by $\mathbb{R}_+$ the set of positive real numbers. Let $\boldsymbol{b}$ be a general vector in $\mathbb{R}_+^K$, $\boldsymbol{a}'$ be a general vector in $\mathbb{R}_+^\kappa$ and $\boldsymbol{a} := (\boldsymbol{a}'; \mathbf{0}_{K-\kappa}) \in \mathbb{R}^K$. Given, the constraints on $\mathbf{\Sigma}_1$ and $\mathbf{\Sigma}_2$, we can parametrize the diagonal elements of $\mathbf{\Sigma}_1^{\frac{1}{2}}$ and $\mathbf{\Sigma}_2^{\frac{1}{2}}$ by $\boldsymbol{a}$ and

$\boldsymbol{b}$, that is, we set $\boldsymbol{\Sigma}_1^{\frac{1}{2}} = \mathrm{diag}(\boldsymbol{a})$ and $\boldsymbol{\Sigma}_2^{\frac{1}{2}} = \mathrm{diag}(\boldsymbol{b})$. The objective function of (32) can then be written as

$$\eta\sqrt{\|\boldsymbol{\mu_1}\|_2^2 + \|\boldsymbol{a} - \mathbf{1}_K\|_2^2} + (1 - \eta)\sqrt{\|\boldsymbol{\mu_2}\|_2^2 + \|\boldsymbol{b} - \mathbf{1}_K\|_2^2}.$$

Combining this last expression and the same colinearity argument as in §D.2 (supported by Fig. 7), (32) is equivalent to

$$\min_{\substack{\boldsymbol{\mu}_1, \boldsymbol{\mu}_2 \in \mathbb{R}^K, \, \boldsymbol{b} \in \mathbb{R}_+^K, \, \boldsymbol{a}' \in \mathbb{R}_+^\kappa, \, \boldsymbol{a}=(\boldsymbol{a}';\boldsymbol{0}_{K-\kappa}), \\ (\boldsymbol{\mu}_1;\boldsymbol{a}),(\boldsymbol{\mu}_2;\boldsymbol{b}),(\boldsymbol{0}_K;\mathbf{1}_K) \text{ are colinear} \\ \& \|\boldsymbol{\mu}_1 - \boldsymbol{\mu}_2\|_2 = \epsilon}} \eta \left\|(\boldsymbol{\mu}_1; \boldsymbol{a}) - (\boldsymbol{0}_K; \mathbf{1}_K)\right\|_2 + (1 - \eta)\left\|(\boldsymbol{\mu}_2; \boldsymbol{b}) - (\boldsymbol{0}_K; \mathbf{1}_K)\right\|_2.$$

$$(33)$$

**Step III:** We solve (33). By the colinearity constraint, we can write $(\boldsymbol{0}_K; \mathbf{1}_K) = u(\boldsymbol{\mu}_2; \boldsymbol{b}) - (u - 1)(\boldsymbol{\mu}_1; \boldsymbol{a})$, where $u \in \mathbb{R}$. We thus obtain that

$$(\boldsymbol{\mu}_2; \boldsymbol{b}) - (\boldsymbol{0}_K; \mathbf{1}_K) = (u - 1)\left((\boldsymbol{\mu}_1; \boldsymbol{a}) - (\boldsymbol{\mu}_2; \boldsymbol{b})\right)$$
$$(\boldsymbol{\mu}_1; \boldsymbol{a}) - (\boldsymbol{0}_K; \mathbf{1}_K) = u\left((\boldsymbol{\mu}_1; \boldsymbol{a}) - (\boldsymbol{\mu}_2; \boldsymbol{b})\right). \tag{34}$$

Furthermore, denoting the coordinates of $\boldsymbol{a}'$ and $\boldsymbol{b}$ by $\{a_i\}_{i=1}^\kappa$ and $\{b_i\}_{i=1}^K$, we similarly obtain that

$$\boldsymbol{0}_K = u\boldsymbol{\mu}_2 - (u - 1)\boldsymbol{\mu}_1$$
$$1 = ub_i - (u - 1)a_i, \quad 1 \le i \le \kappa \tag{35}$$
$$1 = ub_i, \quad d + 1 \le i \le K$$

The last two of equations imply that

$$\sum_{i=1}^\kappa (a_i - b_i)^2 = \frac{\|\mathbf{1}_\kappa - \boldsymbol{a}'\|_2^2}{u^2}$$

and

$$\sum_{i=\kappa+1}^K b_i^2 = \frac{K - \kappa}{u^2}.$$

Combining (30), (34) and the above two equations, we rewrite the objective function of (33) as follows:

$$(\eta|u| + |u - 1|(1 - \eta))\left\|(\boldsymbol{\mu}_1; \boldsymbol{a}) - (\boldsymbol{\mu}_2; \boldsymbol{b})\right\|_2$$

$$= (\eta|u| + |u - 1|(1 - \eta))\sqrt{\|\boldsymbol{\mu}_1 - \boldsymbol{\mu}_2\|_2^2 + \sum_{i=1}^\kappa (a_i - b_i)^2 + \sum_{i=\kappa+1}^K b_i^2}$$

$$\ge (\eta|u| + |u - 1|(1 - \eta))\sqrt{\epsilon^2 + \frac{\|\mathbf{1}_\kappa - \boldsymbol{a}'\|_2^2}{u^2} + \frac{K - \kappa}{u^2}} \tag{36}$$

$$= \left\{(K - \kappa)\left((1 - \eta)\left|\frac{u - 1}{u}\right| + \eta\right)^2 + \epsilon^2\left(\eta|u| + |u - 1|(1 - \eta)\right)^2 \right.$$

$$\left. + \|\mathbf{1}_\kappa - \boldsymbol{a}'\|_2^2\left((1 - \eta)\left|\frac{u - 1}{u}\right| + \eta\right)^2\right\}^{1/2},$$

where equality is achieved if and only if $\|\boldsymbol{\mu}_1 - \boldsymbol{\mu}_2\|_2 = \epsilon$. One can make the following two observations: $u = 0$ does not yield a minimizer of (33), and for any $u \ne 0$, (36) obtains its minimum at $\boldsymbol{a}' = \mathbf{1}_\kappa$. In view of these observations and the derivation above, we define

$$f(u) := (K - \kappa)\left((1 - \eta)\left|\frac{u - 1}{u}\right| + \eta\right)^2 + \epsilon^2\left(\eta|u| + |u - 1|(1 - \eta)\right)^2, \tag{37}$$

and note that (33) is equivalent to

$$\min_{u \ne 0} \quad \sqrt{f(u)}. \tag{38}$$

We rewrite $f(u)$ as

$$f(u) = \begin{cases} (K - \kappa)\left(\frac{u-1}{u}(1 - \eta) + \eta\right)^2 + \epsilon^2\left(\eta u + (1 - \eta)(u - 1)\right)^2, & u \ge 1 \\ (K - \kappa)\left(\frac{1-u}{u}(1 - \eta) + \eta\right)^2 + \epsilon^2\left(\eta u + (1 - \eta)(1 - u)\right)^2, & 1 \ge u > 0 \\ (K - \kappa)\left(\frac{u-1}{u}(1 - \eta) + \eta\right)^2 + \epsilon^2\left(\eta u + (1 - \eta)(u - 1)\right)^2, & 0 > u \end{cases}$$

We denote

$$r_1(u) := (K - \kappa)\left(\frac{u - 1}{u}(1 - \eta) + \eta\right)^2 + \epsilon^2\left(\eta u + (1 - \eta)(u - 1)\right)^2$$

and

$$r_2(u) := (K - \kappa) \left( \frac{1-u}{u}(1-\eta) + \eta \right)^2 + \epsilon^2 \left( \eta u + (1-\eta)(1-u) \right)^2.$$

Their derivatives are

$$r_1'(u) = \frac{2}{u^3} \left( u - (1-\eta) \right) \left( \epsilon^2 u^3 + (K-\kappa)(1-\eta) \right)$$

and

$$r_2'(u) = \frac{2}{u^3} \left( (2\eta - 1)u + (1-\eta) \right) \left( \epsilon^2 (2\eta - 1)u^3 - (K-\kappa)(1-\eta) \right).$$

These expressions for $r_1'$ and $r_2'$ imply that the critical points for $r_1$ are

$$u_{r_1}^{(1)} = 1 - \eta \quad \text{and} \quad u_{r_1}^{(2)} = - \left( \frac{(K-\kappa)(1-\eta)}{\epsilon^2} \right)^{\frac{1}{3}}$$

and the critical points for $r_2$ are

$$u_{r_2}^{(1)} = - \left( \frac{1-\eta}{2\eta - 1} \right) \quad \text{and} \quad u_{r_2}^{(2)} = \left( \frac{(K-\kappa)(1-\eta)}{\epsilon^2 (2\eta - 1)} \right)^{\frac{1}{3}}.$$

We note that $r_1$ is increasing on $(u_{r_1}^{(2)}, 0) \cup (u_{r_1}^{(1)}, \infty)$ and decreasing on $(-\infty, u_{r_1}^{(2)}) \cup (0, u_{r_1}^{(1)})$. On the other hand, $r_2$ is increasing on $(u_{r_2}^{(1)}, 0) \cup (u_{r_2}^{(2)}, \infty)$ and decreasing on $(-\infty, u_{r_2}^{(1)}) \cup (0, u_{r_2}^{(2)})$. Since $\eta > \eta^\star = \frac{K-\kappa+\epsilon^2}{K-\kappa+2\epsilon^2}$, $u_{r_2}^{(2)} \in (0, 1)$. The derivative of $f$ with respect to $u$ is

$$f_u'(u) = \begin{cases} r_1'(u), & u > 0 \\ r_2'(u), & 1 > u > 0 \\ r_1'(u), & 0 > u. \end{cases}$$

So $f(\cdot)$ is increasing on $(u_{r_1}^{(2)}, 0) \cup (u_{r_2}^{(2)}, \infty)$ and decreasing on $(-\infty, u_{r_1}^{(2)}) \cup (0, u_{r_2}^{(2)})$. The values of $f$ at $u_{r_2}^{(2)}$ and $u_{r_1}^{(2)}$ are

$$f(u_{r_2}^{(2)}) = \left( \left( \frac{(K-\kappa)(1-\eta)(2\eta-1)^2}{\epsilon^2} \right)^{\frac{1}{3}} + (1-\eta) \right)^2 \left( (K-\kappa)^{\frac{1}{3}} \left( \frac{\epsilon^2(2\eta-1)}{(1-\eta)} \right)^{\frac{2}{3}} + \epsilon^2 \right),$$

$$f(u_{r_1}^{(2)}) = \left( \left( \frac{(K-\kappa)(1-\eta)}{\epsilon^2} \right)^{\frac{1}{3}} + (1-\eta) \right)^2 \left( (K-\kappa)^{\frac{1}{3}} \left( \frac{\epsilon^2}{(1-\eta)} \right)^{\frac{2}{3}} + \epsilon^2 \right).$$

Consequently, the minimum of $f$ is obtained at $u^\star := u_{r_2}^{(2)}$. By (34) and (35), the means $\boldsymbol{\mu}_1$, $\boldsymbol{\mu}_2$ and the covariance matrices $\boldsymbol{\Sigma}_1$, $\boldsymbol{\Sigma}_2$ satisfy: $\mathbf{0}_K = u^\star \boldsymbol{\mu_2} + (1-u^\star)\boldsymbol{\mu_1}$, $\boldsymbol{\Sigma}_1 = \mathrm{diag}(\mathbf{1}_\kappa; \mathbf{0}_{K-\kappa})$ and $\boldsymbol{\Sigma}_2 = \mathrm{diag}(\mathbf{1}_\kappa; (u^\star)^{-2}\mathbf{1}_{K-\kappa})$. Moreover, the norms of $\boldsymbol{\mu}_1$ and $\boldsymbol{\mu}_2$ can be computed from (35) as $u^\star \epsilon$ and $(1-u^\star)\epsilon$, respectively.

### D.5 PROOF OF PROPOSITION D.2

Notice that since $\boldsymbol{\Sigma}_0 \in \mathcal{S}_{++}^K$, $\det(\boldsymbol{\Sigma}_0) > 0$. On the other hand, since $\boldsymbol{\Sigma}_1 \in \mathcal{S}_+^K$ with $\mathrm{rank}(\boldsymbol{\Sigma}_1) = \kappa < K$, $\det(\boldsymbol{\Sigma}_1) = 0$. Therefore,

$$\log \frac{\det(\boldsymbol{\Sigma}_0)}{\det(\boldsymbol{\Sigma}_1)} = \log \det(\boldsymbol{\Sigma}_0) - \log \det(\boldsymbol{\Sigma}_1) = \infty$$

and this observation and (24) imply that $KL(\mathcal{N}(\boldsymbol{\mu}_1, \boldsymbol{\Sigma}_1) \| \mathcal{N}(\boldsymbol{\mu}_0, \boldsymbol{\Sigma}_0)) = \infty$.

## E ADDITIONAL DETAILS ON THE BENCHMARK METHODS

We overview the benchmark methods compared with MAW, where we present them according to alphabetical order of names. We will include all tested codes in a supplemental webpage.

For completeness, we mention the following links (or papers with links) we used for the different codes. For DSEBMs and DAGMM we used the codes of Golan & El-Yaniv (2018). For LOF, OCSVM and IF we used the scikit-learn (Buitinck et al., 2013) packages for novelty detection. For OCGAN we used its TensorFlow implementation from `https://pypi.org/project/ocgan`. For RSRAE, we adapted the code of Lai et al. (2020) to novelty detection.

All experiments were executed on a Linux machine with 64GB RAM and four GTX1080Ti GPUs.

We remark that for the neural networks based methods (DAGMM, DSEBMs, OCGAN and RSRAE), we followed similar implementation details as the one described in §A.2 for MAW.

**Deep Autoencoding Gaussian Mixture Model (DAGMM)** Zong et al. (2018) is a deep autoencoder model. It optimizes an end-to-end structure that contains both an autoencoder and an estimator for a Gaussian mixture model. Anomalies are detected using this Gaussian mixture model. We remark that this mixture model is proposed for the inliers.

**Deep Structured Energy-Based Models (DSEBMs)** Zhai et al. (2016) makes decision based on an energy function which is the negative log probability that a sample follows the data distribution. The energy based model is connected to an autoencoder in order to avoid the need of complex sampling methods.

**Isolation Forest (IF)** Liu et al. (2008) iteratively constructs special binary trees for the training set and identifies anomalies in the test set as the ones with short average path lengths in the trees.

**Local Outlier Factor (LOF)** Breunig et al. (2000) measures how isolated a data point is from its surrounding neighborhood. This measure is based on an estimation of the local density of a data point using its $k$ nearest neighbors. In the novelty detection setting, it identifies novelties according to low density regions learned from the training data.

**One-class Novelty Detection Using GANs (OCGAN)** Perera et al. (2019) is composed of four neural networks: a denoising autoencoder, two adversarial discriminators, and a classifier. It aims to adversarially push the autoencoder to learn only the inlier features.

**One-Class SVM (OCSVM)** Heller et al. (2003) estimates the margin of the training set, which is used as the decision boundary for the test set. Usually it utilizes a radial basis function kernel to obtain flexibility.

**Robust Subspace Recovery Autoencoder (RSRAE)** Lai et al. (2020) uses an autoencoder structure together with a linear RSR layer imposed with a penalty based on the $\ell_{2,1}$ energy. The RSR layer extracts features of inliers in the latent code while helping to reject outliers. The instances with higher reconstruction errors are viewed as outliers. RSRAE trains a model using the training data. We then apply this model for detecting novelties in the test data.

## F    ADDITIONAL DETAILS ON THE DIFFERENT DATASETS

Below we provide additional details on the five datasets used in our experiments. We remark that each dataset contains several clusters (2 for KDDCUP-99, 5 for Reuters-21578, 3 for COVID-19, 11 largest ones for Caltech101 and 10 for Fashion MNIST). We summarize the number of inliers and outliers per dataset (for both training and testing) in Table 1.

**KDDCUP-99** is a classic dataset for intrusion detection. It contains feature vectors of connections between internet protocols and a binary label for each feature vector identifying normal vs. abnormal ones. The abnormal ones are associated with an "attack" or "intrusion".

**Reuters-21578** contains 21,578 documents with 90 text categories having multi-labels. Following Lai et al. (2020), we consider the five largest classes with single labels. We utilize the scikit-learn packages: TFIDF and Hashing Vectorizer (Rajaraman & Ullman, 2011) to preprocess the documents into 26,147 dimensional vectors.

**COVID-19 (Radiography)** contains chest X-ray RGB images, which are labeled according to the following three categories: COVID-19 positive, normal and bacterial Pneumonia cases. We resize the images to size $64 \times 64$ and rescale the pixel intensities to lie in $[-1, 1]$.

**Caltech101** contains RGB images of objects from 101 categories with identifying labels. Following Lai et al. (2020) we use the largest 11 classes and preprocess their images to have size $32 \times 32$ and rescale the pixel intensities to lie in $[-1, 1]$.

**Fashion MNIST** is an image dataset containing 10 categories of grayscale images of clothing and accessories items. Each image is of size $28 \times 28$ and we rescale the pixel intensities to lie in $[-1, 1]$

We remark that COVID-19, Caltech101 and Reuters-21578 separate between training and testing datapoints. For KDDCUP-99, we randomly split it into training and testing datasets of equal sizes.

Table 1: Numbers of inliers and outliers for training and testing used in the five datasets.

| | Training | | Testing | |
|---|---|---|---|---|
| Datasets | #Inliers ($N$) | #Outliers ($N \times c$) | #Inliers ($N_{\text{test}}$) | #Outliers ($N_{\text{test}} \times c_{\text{test}}$) |
| KDDCUP-99 | 6000 | 6000 $\times c$ | 1200 | 1200 $\times c_{\text{test}}$ |
| Reuters-21578 | 350 | 350 $\times c$ | 140 | 140 $\times c_{\text{test}}$ |
| COVID-19 (Radiography) | 160 | 160 $\times c$ | 60 | 60 $\times c_{\text{test}}$ |
| Caltech101 | 100 | 100 $\times c$ | 100 | 100 $\times c_{\text{test}}$ |
| Fashion MNIST | 300 | 300 $\times c$ | 60 | 60 $\times c_{\text{test}}$ |

## G   EXPERIMENTS WITH DIFFERENT OUTLIER TYPES

In this section, we test the performance of MAW and the benchmark methods when the training and test sets are corrupted by outliers with different structures. We generate a dataset, which we call "Mix Caltech101", in the following way. We fix the largest class of Caltech101 (containing airplane images) as the inlier class and randomly split it into the training inlier class (68.75 %) and testing inlier class (31.25 %). We form the training set by corrupting the training inlier class with random samples from the ten classes of CIFAR10 (Krizhevsky et al., 2009) with training outlier ratio $c \in \{0.1, 0.2, 0.3, 0.4, 0.5\}$. For the test set, we corrupt the testing inlier class by "tile images" from MVTech dataset (Bergmann et al., 2019) with testing outlier ratio $c_{\text{test}}$ in $\{0.1, 0.3, 0.5, 0.7, 0.9\}$. The rest of the settings of the experiments are identical to the description in §4.2. We present the AUC and AP scores and their standard deviations in Fig. 8.

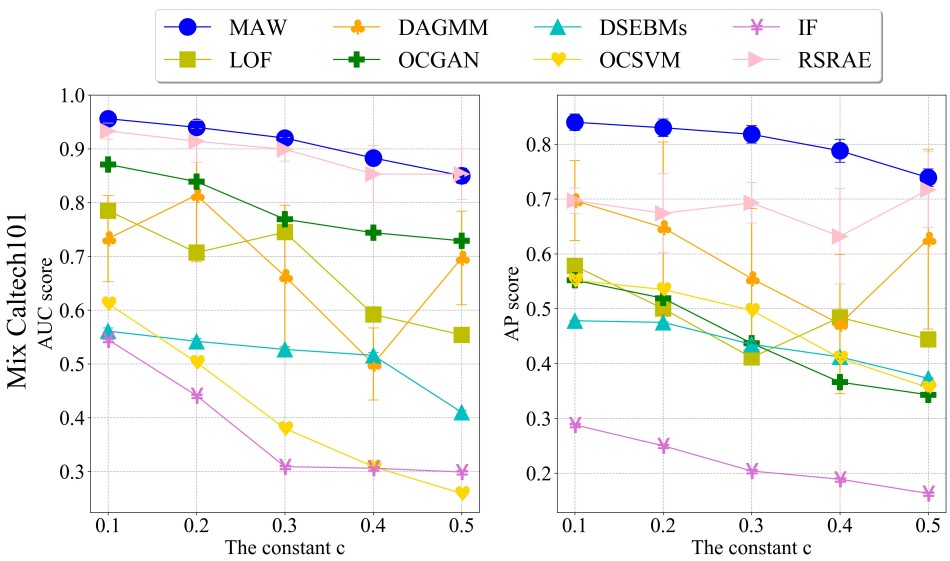

Figure 8: AUC and AP scores with training outlier ratio $c \in \{0.1, 0.2, 0.3, 0.4, 0.5\}$ for the Mix Caltech101 dataset.

The competitive advantage of MAW in comparison to the rest of the methods is also noticeable in this setting. We note that OCSVM, the traditional distance-based method, and IF, the traditional density-based method, perform poorly in this scenario, whereas they performed well in our original setting.

## H   NUMERICAL RESULTS OF EXPERIMENTS

We present as tables the numerical values depicted in Figs. 2 and 3 in §H.1 and those in Fig. 4 in §H.2.

## H.1 TABLE REPRESENTATION FOR FIGS. 2 AND 3

Tables 2-11 report the averaged AUC and AP scores with training outlier ratios $c \in \{0.1, 0.2, 0.3, 0.4, 0.5\}$ that were depicted in Figs. 2 and 3. Each table describes one of the averaged scores (AUC or AP) for one of the five datasets (KDDCUP-99, Reuters-21578, COVID-19, Caltech101 and Fashion MNIST) and also indicates the standard deviation of each value. The outperforming methods are marked in bold.

Table 2: AUC scores of KDDCUP-99.

| Methods | Training outlier ratios c | | | | |
| --- | --- | --- | --- | --- | --- |
| | 0.1 | 0.2 | 0.3 | 0.4 | 0.5 |
| MAW | $\mathbf{0.935} \pm 0.028$ | $\mathbf{0.888} \pm 0.026$ | $\mathbf{0.832} \pm 0.016$ | $\mathbf{0.764} \pm 0.023$ | $0.724 \pm 0.012$ |
| DAGMM | $0.614 \pm 0.083$ | $0.660 \pm 0.109$ | $0.584 \pm 0.133$ | $0.457 \pm 0.099$ | $0.521 \pm 0.089$ |
| DSEBMs | $0.514 \pm 0.000$ | $0.499 \pm 0.000$ | $0.497 \pm 0.000$ | $0.496 \pm 0.000$ | $0.496 \pm 0.000$ |
| IF | 0.811 | 0.527 | 0.516 | 0.750 | 0.706 |
| LOF | 0.480 | 0.527 | 0.516 | 0.527 | 0.530 |
| OCGAN | $0.651 \pm 0.157$ | $0.552 \pm 0.157$ | $0.617 \pm 0.191$ | $0.517 \pm 0.146$ | $0.628 \pm 0.155$ |
| OCSVM | 0.502 | 0.568 | 0.567 | 0.555 | 0.534 |
| RSRAE | $0.815 \pm 0.031$ | $0.839 \pm 0.059$ | $0.778 \pm 0.086$ | $0.735 \pm 0.066$ | $\mathbf{0.740} \pm 0.056$ |

Table 3: AP scores of KDDCUP-99.

| Methods | Training outlier ratios c | | | | |
| --- | --- | --- | --- | --- | --- |
| | 0.1 | 0.2 | 0.3 | 0.4 | 0.5 |
| MAW | $\mathbf{0.740} \pm 0.025$ | $\mathbf{0.698} \pm 0.033$ | $\mathbf{0.647} \pm 0.012$ | $\mathbf{0.594} \pm 0.014$ | $0.556 \pm 0.008$ |
| DAGMM | $0.446 \pm 0.047$ | $0.506 \pm 0.064$ | $0.459 \pm 0.087$ | $0.373 \pm 0.109$ | $0.464 \pm 0.998$ |
| DSEBMs | $0.450 \pm 0.000$ | $0.447 \pm 0.000$ | $0.446 \pm 0.000$ | $0.444 \pm 0.000$ | $0.444 \pm 0.000$ |
| IF | 0.636 | 0.6331 | 0.562 | 0.493 | 0.457 |
| LOF | 0.391 | 0.407 | 0.392 | 0.394 | 0.391 |
| OCGAN | $0.582 \pm 0.132$ | $0.472 \pm 0.163$ | $0.525 \pm 0.133$ | $0.418 \pm 0.136$ | $0.535 \pm 0.133$ |
| OCSVM | 0.543 | 0.598 | 0.595 | 0.438 | 0.426 |
| RSRAE | $0.704 \pm 0.048$ | $0.698 \pm 0.050$ | $0.606 \pm 0.065$ | $0.584 \pm 0.034$ | $\mathbf{0.574} \pm 0.046$ |

Table 4: AUC scores of Reuters-21578.

| Methods | Training outlier ratios c | | | | |
| --- | --- | --- | --- | --- | --- |
| | 0.1 | 0.2 | 0.3 | 0.4 | 0.5 |
| MAW | $0.885 \pm 0.028$ | $\mathbf{0.830} \pm 0.013$ | $0.770 \pm 0.017$ | $\mathbf{0.700} \pm 0.002$ | $\mathbf{0.648} \pm 0.016$ |
| DAGMM | $0.500 \pm 0.000$ | $0.511 \pm 0.027$ | $0.566 \pm 0.110$ | $0.559 \pm 0.087$ | $0.570 \pm 0.091$ |
| DSEBMs | $\mathbf{0.887} \pm 0.012$ | $0.825 \pm 0.012$ | $\mathbf{0.790} \pm 0.015$ | $0.690 \pm 0.002$ | $0.648 \pm 0.010$ |
| IF | 0.544 | 0.535 | 0.520 | 0.453 | 0.452 |
| LOF | 0.757 | 0.612 | 0.579 | 0.631 | 0.616 |
| OCGAN | $0.648 \pm 0.127$ | $0.477 \pm 0.129$ | $0.498 \pm 0.140$ | $0.519 \pm 0.132$ | $0.502 \pm 0.099$ |
| OCSVM | 0.882 | 0.817 | 0.785 | 0.673 | 0.640 |
| RSRAE | $0.786 \pm 0.042$ | $0.755 \pm 0.034$ | $0.716 \pm 0.033$ | $0.605 \pm 0.001$ | $0.494 \pm 0.004$ |

Table 5: AP scores of Reuters-21578.

| Methods | Training outlier ratios c | | | | |
| | 0.1 | 0.2 | 0.3 | 0.4 | 0.5 |
| --- | --- | --- | --- | --- | --- |
| MAW | $0.755 \pm 0.041$ | $0.677 \pm 0.026$ | $0.627 \pm 0.029$ | $\textbf{0.518} \pm 0.004$ | $0.474 \pm 0.013$ |
| DAGMM | $0.316 \pm 0.000$ | $0.316 \pm 0.013$ | $0.365 \pm 0.020$ | $0.362 \pm 0.015$ | $0.372 \pm 0.012$ |
| DSEBMs | $\textbf{0.763} \pm 0.012$ | $\textbf{0.697} \pm 0.011$ | $\textbf{0.666} \pm 0.007$ | $0.515 \pm 0.003$ | $0.473 \pm 0.003$ |
| IF | 0.368 | 0.372 | 0.365 | 0.301 | 0.298 |
| LOF | 0.580 | 0.438 | 0.421 | 0.498 | **0.486** |
| OCGAN | $0.408 \pm 0.045$ | $0.334 \pm 0.098$ | $0.365 \pm 0.106$ | $0.504 \pm 0.083$ | $0.497 \pm 0.094$ |
| OCSVM | 0.746 | 0.681 | 0.637 | 0.467 | 0.438 |
| RSRAE | $0.593 \pm 0.051$ | $0.563 \pm 0.035$ | $0.488 \pm 0.036$ | $0.403 \pm 0.001$ | $0.415 \pm 0.003$ |

Table 6: AUC scores of COVID-19.

| Methods | Training outlier ratios c | | | | |
| | 0.1 | 0.2 | 0.3 | 0.4 | 0.5 |
| --- | --- | --- | --- | --- | --- |
| MAW | $\textbf{0.682} \pm 0.021$ | $\textbf{0.639} \pm 0.018$ | $\textbf{0.606} \pm 0.020$ | $0.551 \pm 0.030$ | $0.534 \pm 0.010$ |
| DAGMM | $0.547 \pm 0.068$ | $0.565 \pm 0.051$ | $0.538 \pm 0.062$ | $0.524 \pm 0.060$ | $0.523 \pm 0.057$ |
| DSEBMs | $0.471 \pm 0.000$ | $0.471 \pm 0.000$ | $0.471 \pm 0.000$ | $0.471 \pm 0.000$ | $0.471 \pm 0.000$ |
| IF | 0.604 | 0.571 | 0.555 | 0.523 | 0.499 |
| LOF | 0.672 | 0.618 | 0.572 | **0.580** | **0.589** |
| OCGAN | $0.492 \pm 0.000$ | $0.492 \pm 0.000$ | $0.492 \pm 0.000$ | $0.485 \pm 0.000$ | $0.491 \pm 0.000$ |
| OCSVM | 0.528 | 0.528 | 0.528 | 0.535 | 0.521 |
| RSRAE | $0.565 \pm 0.031$ | $0.527 \pm 0.028$ | $0.476 \pm 0.023$ | $0.454 \pm 0.018$ | $0.427 \pm 0.011$ |

Table 7: AP scores of COVID-19.

| Methods | Training outlier ratios c | | | | |
| | 0.1 | 0.2 | 0.3 | 0.4 | 0.5 |
| --- | --- | --- | --- | --- | --- |
| MAW | $0.455 \pm 0.014$ | $\textbf{0.442} \pm 0.011$ | $\textbf{0.424} \pm 0.018$ | $0.368 \pm 0.015$ | $0.376 \pm 0.008$ |
| DAGMM | $0.354 \pm 0.053$ | $0.390 \pm 0.057$ | $0.316 \pm 0.052$ | $0.357 \pm 0.050$ | $0.368 \pm 0.047$ |
| DSEBMs | $0.372 \pm 0.000$ | $0.372 \pm 0.000$ | $0.372 \pm 0.000$ | $0.372 \pm 0.000$ | $0.372 \pm 0.000$ |
| IF | 0.425 | 0.404 | 0.392 | 0.380 | 0.373 |
| LOF | **0.463** | 0.422 | 0.402 | **0.397** | **0.393** |
| OCGAN | $0.381 \pm 0.000$ | $0.381 \pm 0.000$ | $0.381 \pm 0.000$ | $0.383 \pm 0.000$ | $0.376 \pm 0.000$ |
| OCSVM | 0.315 | 0.315 | 0.315 | 0.372 | 0.365 |
| RSRAE | $0.388 \pm 0.018$ | $0.377 \pm 0.016$ | $0.355 \pm 0.011$ | $0.352 \pm 0.010$ | $0.340 \pm 0.009$ |

Table 8: AUC scores of Caltech101.

| Methods | Training outlier ratios c | | | | |
| | 0.1 | 0.2 | 0.3 | 0.4 | 0.5 |
| --- | --- | --- | --- | --- | --- |
| MAW | $\textbf{0.801} \pm 0.017$ | $\textbf{0.760} \pm 0.028$ | $\textbf{0.700} \pm 0.038$ | $\textbf{0.608} \pm 0.031$ | $\textbf{0.570} \pm 0.021$ |
| DAGMM | $0.684 \pm 0.100$ | $0.588 \pm 0.115$ | $0.500 \pm 0.100$ | $0.509 \pm 0.101$ | $0.514 \pm 0.095$ |
| DSEBMs | $0.536 \pm 0.011$ | $0.612 \pm 0.025$ | $0.577 \pm 0.030$ | $0.564 \pm 0.021$ | $0.536 \pm 0.021$ |
| IF | 0.755 | 0.694 | 0.626 | 0.575 | 0.540 |
| LOF | 0.674 | 0.593 | 0.495 | 0.436 | 0.411 |
| OCGAN | $0.494 \pm 0.000$ | $0.494 \pm 0.000$ | $0.494 \pm 0.000$ | $0.500 \pm 0.000$ | $0.500 \pm 0.000$ |
| OCSVM | 0.682 | 0.618 | 0.577 | 0.538 | 0.516 |
| RSRAE | $0.774 \pm 0.027$ | $0.722 \pm 0.041$ | $0.664 \pm 0.082$ | $0.579 \pm 0.047$ | $0.568 \pm 0.036$ |

Table 9: AP scores of Caltech101.

| Methods | Training outlier ratios c | | | | |
| | 0.1 | 0.2 | 0.3 | 0.4 | 0.5 |
| --- | --- | --- | --- | --- | --- |
| MAW | **0.634** $\pm$ 0.027 | **0.572** $\pm$ 0.039 | **0.531** $\pm$ 0.064 | 0.412 $\pm$ 0.029 | 0.414 $\pm$ 0.021 |
| DAGMM | 0.574$\pm$ 0.088 | 0.422 $\pm$ 0.112 | 0.308 $\pm$ 0.102 | 0.351 $\pm$ 0.074 | 0.363 $\pm$ 0.076 |
| DSEBMs | 0.385$\pm$ 0.003 | 0.472$\pm$ 0.051 | 0.398$\pm$0.019 | 0.383 $\pm$ 0.023 | 0.365 $\pm$ 0.028 |
| IF | 0.545 | 0.486 | 0.430 | 0.304 | 0.371 |
| LOF | 0.460 | 0.400 | 0.337 | 0.304 | 0.290 |
| OCGAN | 0.362$\pm$ 0.000 | 0.362$\pm$ 0.000 | 0.362 $\pm$ 0.000 | 0.362 $\pm$ 0.000 | 0.362 $\pm$ 0.000 |
| OCSVM | 0.472 | 0.419 | 0.380 | 0.352 | 0.339 |
| RSRAE | 0.595$\pm$ 0.038 | 0.551 $\pm$ 0.045 | 0.495 $\pm$0.073 | **0.425** $\pm$ 0.040 | **0.443** $\pm$ 0.027 |

Table 10: AUC scores of Fashion MNIST

| Methods | Training outlier ratios c | | | | |
| | 0.1 | 0.2 | 0.3 | 0.4 | 0.5 |
| --- | --- | --- | --- | --- | --- |
| MAW | **0.897** $\pm$ 0.013 | **0.879** $\pm$ 0.011 | **0.852** $\pm$ 0.022 | 0.830 $\pm$ 0.017 | 0.801 $\pm$ 0.016 |
| DAGMM | 0.607 $\pm$ 0.093 | 0.376 $\pm$ 0.070 | 0.427 $\pm$ 0.090 | 0.401 $\pm$ 0.078 | 0.411 $\pm$ 0.081 |
| DSEBMs | 0.730 $\pm$ 0.092 | 0.729 $\pm$ 0.105 | 0.739 $\pm$ 0.086 | 0.723 $\pm$ 0.106 | 0.687 $\pm$ 0.096 |
| IF | 0.893 | 0.875 | 0.843 | **0.834** | **0.827** |
| LOF | 0.569 | 0.507 | 0.476 | 0.468 | 0.458 |
| OCGAN | 0.542 $\pm$ 0.006 | 0.538 $\pm$ 0.004 | 0.544 $\pm$ 0.014 | 0.531 $\pm$ 0.003 | 0.525 $\pm$ 0.004 |
| OCSVM | 0.895 | 0.874 | 0.848 | 0.831 | 0.814 |
| RSRAE | 0.860 $\pm$ 0.022 | 0.848 $\pm$ 0.022 | 0.829 $\pm$ 0.042 | 0.831 $\pm$ 0.028 | 0.808 $\pm$ 0.028 |

Table 11: AP scores of Fashion MNIST

| Methods | Training outlier ratios c | | | | |
| | 0.1 | 0.2 | 0.3 | 0.4 | 0.5 |
| --- | --- | --- | --- | --- | --- |
| MAW | **0.788** $\pm$0.013 | 0.754 $\pm$ 0.014 | 0.723$\pm$0.029 | 0.686 $\pm$ 0.025 | 0.672 $\pm$0.021 |
| DAGMM | 0.482 $\pm$0.051 | 0.303 $\pm$0.057 | 0.334 $\pm$0.113 | 0.318 $\pm$0.056 | 0.330 $\pm$ 0.038 |
| DSEBMs | 0.600 $\pm$ 0.045 | 0.609$\pm$ 0.120 | 0.613$\pm$0.089 | 0.605 $\pm$0.086 | 0.565 $\pm$ 0.072 |
| IF | 0.768 | 0.724 | 0.693 | 0.665 | 0.642 |
| LOF | 0.382 | 0.331 | 0.308 | 0.301 | 0.294 |
| OCGAN | 0.504 $\pm$ 0.002 | 0.503 $\pm$ 0.003 | 0.500 $\pm$ 0.059 | 0.495 $\pm$ 0.001 | 0.493 $\pm$ 0.001 |
| OCSVM | 0.801 | **0.768** | **0.735** | **0.696** | 0.664 |
| RSRAE | 0.749 $\pm$ 0.029 | 0.736 $\pm$ 0.032 | 0.716 $\pm$ 0.048 | 0.683 $\pm$ 0.036 | **0.680** $\pm$ 0.042 |

## H.2 TABLE REPRESENTATION FOR FIG. 4

Tables 12-15 record the averaged AUC and AP scores with training outlier ratios $c = 0.1, 0.2, 0.3$ that were depicted in Fig. 4. Each table describes one of the averaged scores (AUC or AP) for one of the two representative datasets (KDDCUP-99 and COVID-19) and also indicates the standard deviation of each value. The outperforming methods are marked in bold.

Table 12: AUC scores of KDD-99 for variations of MAW

| Methods | Training outlier ratios c | | |
|---|---|---|---|
| | 0.1 | 0.2 | 0.3 |
| MAW | **0.936** $\pm$ 0.028 | **0.888** $\pm$ 0.030 | **0.832** $\pm$ 0.016 |
| MAW-MSE | 0.844 $\pm$ 0.039 | 0.812 $\pm$ 0.032 | 0.746 $\pm$ 0.044 |
| MAW-KL divergence | 0.905 $\pm$ 0.026 | 0.863 $\pm$ 0.028 | 0.801 $\pm$ 0.029 |
| MAW-same rank | 0.912 $\pm$ 0.023 | 0.868 $\pm$ 0.011 | 0.797 $\pm$ 0.022 |
| MAW-single Gaussian | 0.914 $\pm$ 0.016 | 0.862 $\pm$ 0.021 | 0.796 $\pm$ 0.013 |
| MAW-diagonal cov. | 0.918 $\pm$ 0.023 | 0.858 $\pm$ 0.020 | 0.801 $\pm$ 0.044 |
| VAE | 0.821 $\pm$ 0.048 | 0.785 $\pm$ 0.027 | 0.732 $\pm$ 0.046 |

Table 13: AP scores of KDDCUP-99 for variations of MAW

| Methods | Training outlier ratios c | | |
|---|---|---|---|
| | 0.1 | 0.2 | 0.3 |
| MAW | **0.740** $\pm$ 0.025 | **0.698** $\pm$ 0.033 | **0.647** $\pm$ 0.012 |
| MAW-MSE | 0.715 $\pm$ 0.079 | 0.589 $\pm$ 0.058 | 0.524 $\pm$ 0.053 |
| MAW-KL divergence | 0.735 $\pm$ 0.028 | 0.676 $\pm$ 0.028 | 0.618 $\pm$ 0.024 |
| MAW-same rank | 0.725 $\pm$ 0.028 | 0.681 $\pm$ 0.015 | 0.622 $\pm$ 0.024 |
| MAW-single Gaussian | 0.737 $\pm$ 0.018 | 0.675 $\pm$ 0.023 | 0.620 $\pm$ 0.025 |
| MAW-diagonal cov. | 0.724 $\pm$ 0.021 | 0.678 $\pm$ 0.035 | 0.589 $\pm$ 0.064 |
| VAE | 0.642 $\pm$ 0.030 | 0.555 $\pm$ 0.043 | 0.524 $\pm$ 0.028 |

Table 14: AUC scores of COVID-19 for variations of MAW

| Methods | Training outlier ratios c | | |
|---|---|---|---|
| | 0.1 | 0.2 | 0.3 |
| MAW | **0.682** $\pm$ 0.021 | **0.639** $\pm$ 0.018 | **0.606** $\pm$ 0.029 |
| MAW-MSE | 0.642 $\pm$ 0.022 | 0.584 $\pm$ 0.063 | 0.548 $\pm$ 0.041 |
| MAW-KL divergence | 0.654 $\pm$ 0.025 | 0.610 $\pm$ 0.026 | 0.528 $\pm$ 0.064 |
| MAW-same rank | 0.654 $\pm$ 0.031 | 0.604 $\pm$ 0.048 | 0.557 $\pm$ 0.044 |
| MAW-single Gaussian | 0.651 $\pm$ 0.027 | 0.616 $\pm$ 0.029 | 0.537 $\pm$ 0.047 |
| MAW-diagonal cov. | 0.630 $\pm$ 0.029 | 0.606 $\pm$ 0.030 | 0.545 $\pm$ 0.035 |
| VAE | 0.629 $\pm$ 0.073 | 0.585 $\pm$ 0.065 | 0.532 $\pm$ 0.049 |

Table 15: AP scores of COVID-19 for variations of MAW

| Methods | Training outlier ratios c | | |
|---|---|---|---|
| | 0.1 | 0.2 | 0.3 |
| MAW | **0.459** $\pm$ 0.014 | **0.442** $\pm$ 0.011 | **0.424** $\pm$ 0.018 |
| MAW-MSE | 0.421 $\pm$ 0.015 | 0.395 $\pm$ 0.025 | 0.377 $\pm$ 0.012 |
| MAW-KL divergence | 0.427 $\pm$ 0.016 | 0.403 $\pm$ 0.012 | 0.370 $\pm$ 0.021 |
| MAW-same rank | 0.422 $\pm$ 0.021 | 0.413 $\pm$ 0.026 | 0.375 $\pm$ 0.019 |
| MAW-single Gaussian | 0.425 $\pm$ 0.019 | 0.409 $\pm$ 0.012 | 0.374 $\pm$ 0.016 |
| MAW-diagonal cov. | 0.412 $\pm$ 0.016 | 0.397 $\pm$ 0.018 | 0.369 $\pm$ 0.012 |
| VAE | 0.412 $\pm$ 0.030 | 0.411 $\pm$ 0.043 | 0.379 $\pm$ 0.028 |

