# OpenReview forum: "Novelty Detection via Robust Variational Autoencoding"
_ICLR.cc/2021/Conference — Reject_

### Official Review · AnonReviewer1 · 2020-10-26
**Interesting problem, but the method is not convincing**

**Rating:** 4
**Confidence:** 4

**Review:**

The paper seeks to address the problem of novelty detection under the circumstance of having high corruptions in the training data. This is different from most previous work, which often assumes that training dataset is pure. To address this issue, a VAE-based approach is adopted in this paper, with several modifications made to the vanilla VAE to promote the robustness of VAE in detecting outliers in the corruption circumstance. Among the modifications, the paper assumes the posterior is approximated by a two-component Gaussian mixture distribution, with each having a low-rank and full-rank covariance matrix, respectively. The paper hopes that the posterior of inliers (normal data points) can be represented by the low-rank covariance matrix, while that of outliers cannot. Another notable modification is that the Wasserstein-1 regularization is used to replace the KL-regularization in the ELBO, which is claimed to be more suitable to the low-rank modeling. Some experiments are conducted to evaluate the outlier detection performance of the proposed method under corrupted circumstance.

Strength:
1. The problem of detecting outliers under highly corrupted environment is of practical importance and is not investigated under the extremely corrupted circumstance.

2. The idea of proposing to use low-rank and full-rank geometry characteristics to separate inliers and outliers is interesting.


Weakness:
1. Although the idea of using low/full-rank geometric characteristic to detect outliers is interesting, the paper barely states how to implement this idea, that is, how to enforce the inliers resides in the low-rank covariance matrix, while the outlier will not. Without any specific design, we cannot believe this will be realized automatically.

2. The theoretical result (Section 3) established to argue the superiority of Wasserstein regularization in robustness is better than the original KL regularization in vanilla VAE is not convincing, or is not practically meaningful. That is because the result only reveals that under a very special case, the Wasserstein may learn the true distribution under corrupted environment. We cannot see this result has any implication under a broader or more general circumstance.

3. I guess the true reason to choose W_1 over KL distance is that because a Gaussian mixture posterior is employed, the KL cannot be evaluated in close-form. By resorting to W_1 distance, the distance can be estimated with the samples, but the KL distance cannot be estimated in this way. Overall, I cannot buy the argument that the authors proposed to use the W_1 distance/regularization here.

4. I also have some concerns over the choosing of hyper-parameters. The paper sets the dimension of latent representation (d) to be 2, which, I think, is too small. But if it is set to be an appropriate value (e.g. 100), the posterior with a full-rank covariance matrix will be computationally expensive. Moreover, I also think that it is not a good idea to set the mixture coefficient $\eta$ to be a fixed value. Generally, this should be learned from data, because you cannot know the ratio of different components in advance.

5. I also have some doubts over the experiment settings. At the training stage, the training data is corrupted by a fraction of outliers. But if the data is corrupted by ‘outliers’, that means the model has already made use of the information of outliers. So, if the ratio of outlier in the training dataset is higher, it may be more favorable to the proposed method. For example, if the ratio is large, e.g. c=0.3, this problem under this setting is more like a clustering problem. So, this kind of settings may be not fair to the comparing methods. Maybe, the better setting should be let the training data corrupted by some data that is not in both the training and testing datasets.

---

> ### Author Response · Authors · 2020-11-25
> **Response to Review #1 (part 1)**
>
> Q1. Although the idea of using low/full-rank geometric characteristic to detect outliers is interesting, the paper barely states how to implement this idea, that is, how to enforce the inliers resides in the low-rank covariance matrix, while the outlier will not. Without any specific design, we cannot believe this will be realized automatically.
>
> Response: Since we do not have labels for the training set, we cannot supervisedly learn the two components of the mixture model. However, the use of two robust losses (least absolute deviation and the $W_1$ distance) helps obtain a careful model for the inliers, which is robust to outliers. Note that in our testing we only use the model for the inliers. So the special “design” is the use of robust losses and we cannot foresee any other way of “enforcement” without having labels for the training set.
>
> In order to further explain some intuition of the mechanism of MAW, let us assume that the inliers are sampled from a distribution on a low-dimensional manifold that can be encoded by a Gaussian on a low-dimensional latent space by using a certain mapping. We assume further that the outliers are arbitrary, but the percentage of any subgroup of them, that can be sampled from a similar kind of a distribution, is smaller than that of the inliers. Given this assumption and considering the latent space, MAW aims to model the mixture component of the inliers as a Gaussian with low-rank covariance (and that of the outliers as a Gaussian with full-rank covariance). In order to provide some technical intuition why this happens in practice and how the robust functions help with this, we consider three complementary cases, where either the inlier model has a full rank covariance or the outlier component has a low-rank covariance and we explain why they are unlikely to occur. In the first case, the inliers and outliers are both modeled (in the latent space) by Gaussian distributions with low-rank covariances. In this case, the W1 distance is minimized over a smaller set and thus the loss is increased. We remark that one may not use the KL divergence instead of W1 as we show in our theory that the KL divergence cannot be applied to our low-rank modeling paradigm (see ill-posedness in Proposition D.2 and its proof).  In the second case, both the inliers and outliers are modeled (in the latent space) with full-rank Gaussians. In this case it is most likely that the minimizer for the inliers will be full-rank, and thus due to the assumed low-dimensional structure of the inliers, it will result in an increase of the reconstruction error. In the third case, the inliers are modeled (in the latent space) by a Gaussian with full-rank covariance and the outliers are modeled (in the latent space) by a Gaussian with a low-rank covariance. In this case, the L1-distance in the reconstruction loss will increase the reconstruction loss. The increase due to the L1-distance is more significant than due to the L2-distance. We hope that these more technical ideas help explain why there is a hidden mechanism that makes things work.
>
> Q2. The theoretical result (Section 3) established to argue the superiority of Wasserstein regularization in robustness is better than the original KL regularization in vanilla VAE is not convincing, or is not practically meaningful. That is because the result only reveals that under a very special case, the Wasserstein may learn the true distribution under corrupted environment. We cannot see this result has any implication under a broader or more general circumstance.
>
> Response: We don’t agree that our theory is not practically meaningful. It is a common practice by researchers who care about theory to formulate a special setting under which some guarantees can be provided for a special setting. Such guarantees may highlight some interesting features of the complicated process. In our case, the interesting observations gained by the theory is that minimizing the Wasserstein distance is more robust than minimizing the KL-divergence. Furthermore, the theory shows that the KL divergence is not suitable for a low-rank covariance of any of the mixture components.
>
> We are not sure what is the main objection to our model and theory. Nevertheless, we comment that our epsilon-separation assumption just assures that the two means of the inlier Gaussian and the outlier Gaussian cannot be too close together. Since it may seem restrictive, we replaced it with the equivalent constraint that the distance of the two means is greater than or equal to epsilon (instead of exactly equal to epsilon). Anyway, it is truly challenging to further improve our theory. The current proofs are nontrivial and required some effort and we are sorry that such an effort was not appreciated by the reviewer. We are not aware of a stronger theory in this specific area.

---

> ### Author Response · Authors · 2020-11-25
> **Response to Review #1 (part 2)**
>
> Q3. I guess the true reason to choose W_1 over KL distance is that because a Gaussian mixture posterior is employed, the KL cannot be evaluated in close-form. By resorting to W_1 distance, the distance can be estimated with the samples, but the KL distance cannot be estimated in this way. Overall, I cannot buy the argument that the authors proposed to use the W_1 distance/regularization here.
>
> Response: We do not agree that the KL divergence cannot be implemented in our setting and we never claimed this in the manuscript. One can implement the KL divergence without a closed form expression. Indeed, this can be done by replacing the WGAN in MAW by a regular GAN. Note that such implementation estimates the KL divergence using the samples as opposed to the claim of the reviewer. We actually compared MAW with this variant that uses GAN in Fig. 4 in Section 4.3. Clearly, the performance when using GAN is not as good as when using WGAN.
>
> One of the reasons for our choice of the Wasserstein distances over the KL divergence is the need to obtain robustness. As we carefully explained in our response to Q1, robustness is important for ignoring the outliers in the training set when there are no labels that distinguish the inliers and outliers. Our theory in Section 3 establishes such robustness in a special setting and it is thus crucial. It also shows that the KL divergence is generally not robust. The theory also showed that the KL divergence does not fit to our low-rank modeling, unlike the Wasserstein distance. Note that the goal of this theory was to highlight these properties and not to provide full guarantees. It is thus sufficiently interesting to realize these properties even in a special case (which is already hard to prove) and not in full generality.
>
> Q4. I also have some concerns over the choosing of hyper-parameters. The paper sets the dimension of latent representation (d) to be 2, which, I think, is too small. But if it is set to be an appropriate value (e.g. 100), the posterior with a full-rank covariance matrix will be computationally expensive. Moreover, I also think that it is not a good idea to set the mixture coefficient $\eta$ to be a fixed value. Generally, this should be learned from data, because you cannot know the ratio of different components in advance.
>
> Response: In general our model is not too sensitive to the latent dimension as can be seen in Fig. 5 in Appendix C1. This figure also implies that a small value, such as d=2, generally leads to good performance. We are not sure why the reviewer expects 100 to be an appropriate value (please check this figure). Since one does not know the mixture parameter and there are no labels for outliers in order to validate it from the data, we had it fixed. Nevertheless, we demonstrated in Fig. 6 in Appendix C2 the sensitivity of the precision and recall with the change of $\eta$. We believe that values of c up to 0.3 are reasonable for applications and we notice that for this range of values the results are stable with sufficiently large choices of $\eta$. We also recently noted similar results for c = 0.4, 0.5, but we don’t find them necessary, so we did not include them. However, if the reviewers find them necessary we will also include them.

---

> ### Author Response · Authors · 2020-11-25
> **Response to Review #1 (part 3)**
>
> Q5. I also have some doubts over the experiment settings. At the training stage, the training data is corrupted by a fraction of outliers. But if the data is corrupted by ‘outliers’, that means the model has already made use of the information of outliers. So, if the ratio of outlier in the training dataset is higher, it may be more favorable to the proposed method. For example, if the ratio is large, e.g. c=0.3, this problem under this setting is more like a clustering problem. So, this kind of settings may be not fair to the comparing methods. Maybe, the better setting should be let the training data corrupted by some data that is not in both the training and testing datasets.
>
> Response: Our method is based on robust reconstruction of the inliers. During the testing stage, we only use the inlier mixture component to detect novelties. Thus our method does not take any advantage of correct modeling of the outliers. We do not need the outlier mode to fit the outliers well in order for our method to work. If we still implicitly encode information of the outliers,  then one may make a similar argument to other methods.
>
> Actually, it is easier for our model to handle the case in which the training dataset is polluted by a different class of outliers than those in the test data. Indeed, if there are similar kinds of outliers in both the training and test set, then identification of outliers as inliers in the training set will clearly lead to errors in the test set, but this is not the case when the outliers are different in both cases. It also makes sense in practical applications that the training set may share similar kinds of outliers (for example, patients with flu when trying to train to identify patients with COVID-19).
>
> To further address this point, we implemented an experiment following the suggestion by the reviewer. We report it in Appendix G.  In this experiment, MAW performed the best  with a large margin compared with other methods. Possibly the two methods that are disadvantaged in the suggested setting are OCSVM, the traditional distance-based method, and IF, the traditional density-based method.

---

### Official Review · AnonReviewer2 · 2020-10-28
**This paper propose a robust VAE model for novelty detection (semi-supervised anomaly detection), which allows a nontrivial fraction of corrupted samples (outliers) within the training set.**

**Rating:** 6
**Confidence:** 3

**Review:**

Compared with conventional VAE, this paper incorporates the following strategies to improve VAE for novelty detection from corrupted training data. 1) By considering that outliers tend to have more complex structures, this paper assumes inliers and ourliers lie on Gaussian distributions with different ranks, and proposes a Gaussian mixture model for posterior q(z|x) including two component: a low-rank multivariate Gaussian distribution component for modeling inliers and a full-rank multivariate
Gaussian distribution component for outliers.  2) Applying the Wasserstein distance (W distance) between q(z|x) and p(z) instead of the KL divergence in the original VAE. The paper also proves the superiority of using W distance over KL divergence, by showing that the minimized value exists in W distance between a Gaussian mixture distribution q(z|x) and Gaussian prior p(z). 3) 3)For Rrconstruction loss, this model adopts ||x-D(z)||_2 as described in (3) instead of ||x-D(z)||^2_2, which is used in conventional VAE. This practice helps to alleviate the problem that the loss item in conventional VAE being too small when data point deviates from the center of the Gaussian distribution.
The experiments show that proposed method achieves good result among 4 datasets, which domonstrate the effectiveness of the three strategies.
Advantages of this paper: although Gaussian mixture model has been used in previous work of VAE, this model innovatively propose Gaussian components with different ranks for modeling inlier and outliers to improve the performance for training data with corrupted samples. This work also gives a theoretical guarantee for the advantage of W distance minimization in given setting.
Potential drawbacks: Problem setting is confusing. Outliers/novelties are supposed to be rare in the dataset, while the paper assumes that a nontrivial fraction of outliers existing in the data. In addition, more recent work is not compared, such as Ruff, L., Vandermeulen, R. A., Görnitz, N., Binder, A., Müller, E., Müller, K. R., Kloft, M. (2019). Deep semi-supervised anomaly detection. arXiv preprint arXiv:1906.02694

---

> ### Author Response · Authors · 2020-11-25
> **Response to Review #2 (part1)**
>
> Q1. Problem setting is confusing. Outliers/novelties are supposed to be rare in the dataset, while the paper assumes that a nontrivial fraction of outliers existing in the data.
>
> Response: When having sufficient experience and expertise on a carefully studied area and sufficiently precise tools to collect data, then one can produce training sets with few outliers. However, there are at least two important scenarios, where our method is needed. The first scenario includes new areas of studies, where it is unclear how to distinguish between normal and abnormal points.  For example, in the beginning of the COVID-19 pandemic it was hard to diagnose COVID-19 patients and distinguish them from the rest of patients, for example, patients with flu. Therefore, if you focus on the set of COVID-19 positive patients and designate them as inliers, then you can expect a nontrivial portion of outliers, among the admitted patients.
>
> Another scenario occurs when it is very hard to make precise measurements. This scenario occurs in some image processing/computer vision tasks of cryogenic electron microscopy (cryo-EM), which aims at reconstructing 3D structures of biological molecules at near-atomic resolution from their 2D projection images. Such reconstruction needs to start with the removal of outliers. However, in practice, there might be a huge portion of 2D images that are outliers (outlier ratios may up to 40% or more). These anomalous images may come from the damage of the biomolecules themselves or from the unrelated information (e.g. micrographic noise produced by cameras or micro water drop or ice on the particles). Due to the limited methods for outlier detection with large corruption, researchers usually need to manually pick out anomalous images. However, this is laborious and error-prone, since the scale is too micro to be visible.
>
> We thus believe that our setting with non-trivial corruption (e.g. training outlier ratio up to 0.3) is valuable to the practitioner in certain situations. Also note that even in the case where the outlier ratio is as small (e.g., 10%), our method still generally outperforms other benchmarks (see Figs. 2 and 3 for example).

---

> ### Author Response · Authors · 2020-11-25
> **Response to Review #2 (part 2)**
>
> Q2. In addition, more recent work is not compared, such as Ruff, L., Vandermeulen, R. A., Görnitz, N., Binder, A., Müller, E., Müller, K. R., Kloft, M. (2019). Deep semi-supervised anomaly detection. arXiv preprint arXiv:1906.02694
>
> Response: We thank the reviewer for pointing out this paper.
> Although the title of the paper is “semi-supervised anomaly detection”, we have to point out that the setting of that paper is different than ours. In their context, “semi-supervised” refers to the availability of a “small” portion of training data for both the inliers and outliers. However, this is not the setting of “novelty detection” and of what is commonly called semi-supervised anomaly detection (here the difference between supervised and semi-supervised is by having an additional training set for the outliers and it is independent of the number of training points, unlike the common use of these terms, e.g., in clustering).
> Nevertheless, since the DeepSAD model can still be used even if there are no outliers for training, we tested DeepSAD under the novelty detection setting and compared it with MAW on the four datasets with a fixed training outlier ratio of 0.3. With the same experiment settings as described in Section 4.2, we present the results of AUC, AP and their standard deviations as tables below:
>
> ===================================================================================================
>
> AUC scores when the training outlier ratio = 0.3
>
> ++++++++++++++++++++++++++++++++++++++++++++++++++++++++++++++++++++++++++++++++++++++++++++++++
>
> Dataset  $\hspace{2cm}$ KDDCUP-99 $\hspace{2cm}$ COVID-19 $\hspace{2cm}$ Caltech101 $\hspace{2cm}$ Reuters-21578
>
> ++++++++++++++++++++++++++++++++++++++++++++++++++++++++++++++++++++++++++++++++++++++++++++++++
>
> MAW  $\hspace{2.3cm}$ 0.832 ± 0.016 $\hspace{1.8cm}$ 0.606 ± 0.020 $\hspace{1.7cm}$ 0.700 ± 0.038  $\hspace{1.7cm}$ 0.770 ± 0.017
>
> DeepSAD $\hspace{1.7cm}$ 0.725 ± 0.000 $\hspace{1.7cm}$ 0.399 ± 0.007 $\hspace{1.7cm}$ 0.434 ± 0.002 $\hspace{1.7cm}$ 0.537 ± 0.012
>
> ===================================================================================================
>
>
> ===================================================================================================
>
> AP scores when the training outlier ratio = 0.3
>
> ++++++++++++++++++++++++++++++++++++++++++++++++++++++++++++++++++++++++++++++++++++++++++++++++
>
> Dataset  $\hspace{2cm}$ KDDCUP-99  $\hspace{2cm}$ COVID-19  $\hspace{2cm}$ Caltech101  $\hspace{2cm}$ Reuters-21578
>
> ++++++++++++++++++++++++++++++++++++++++++++++++++++++++++++++++++++++++++++++++++++++++++++++++
>
> MAW  $\hspace{2.3cm}$ 0.647 ± 0.012 $\hspace{1.8cm}$ 0.424 ± 0.018 $\hspace{1.7cm}$ 0.531 ± 0.064  $\hspace{1.7cm}$ 0.627 ± 0.029
>
> DeepSAD $\hspace{1.7cm}$ 0.407 ± 0.000 $\hspace{1.7cm}$ 0.271 ± 0.002 $\hspace{1.7cm}$ 0.302 ± 0.002 $\hspace{1.7cm}$ 0.373 ± 0.004
>
> ===================================================================================================
>
> We can see that DeepSAD generally is not comparable with MAW. This is not surprising because DeepSAD has a different goal in its design. We feel if we include this result in our paper, it will be unfair (and deeply sad) to DeepSAD. Thus, we choose not to include it as a compared benchmark. Moreover, we compared with DSEBMs, DAGMM, and OCGAN, which are all recent works and attained success in novelty detection.
>
> We emphasize that in the current revised version, we are now citing this work in the introduction part to emphasize the common meaning of “semi-supervised” anomaly detection, while also being aware of another use of this name.

---

### Official Review · AnonReviewer3 · 2020-10-29
**Presenting a robust method to the noisy training dataset for novelty detection by modeling mixture of Gaussian with outlier and inlier distribution in the latent space**

**Rating:** 5
**Confidence:** 4

**Review:**

This study proposes a novel method that can work well even the training data is corrupted by partial data from the unknown domain. Though it deals with the well-known problem called 'Noisy data/label', its approach is not the same thing as the previous works as it focuses on variational autoencoder on the task of novelty detection. And its arguments and statistical assumptions are followed by mathematical proofs.

Overall, it is an interesting approach and I believe it would give a good way to ML practitioners who are struggling with noisy datasets in real-world applications. However, there some questions/comments about the article which may make the study more consolidate:

Questions
- In the description of the proposed method, MAW, Discriminator generates Loss(Lw1) by comparing between Zgen and Zhyp. And Zhyp is unimodal distribution while Zgen follows MoG. I wonder whether there is a risk that inlier and outlier distributions are mixed(combined) as the loss makes the generator generates just the same mu/sigma regardless of the domains. If so, is there any equilibrium trick required so that the generator would not be strong too much?

- Though it is hard to pre-estimate how the outlier distribution looks like, it is more common to assume the outlier distribution has multi-modal than uni-modal. However, the proposed method approximates the outlier distribution as unimodal Gaussian distribution. Is it possible to model the outliers as multi-modal distribution such as MoG?

- In the experiment with the multiclass dataset, the number of possible inlier domains is the same as the number of classes in the dataset. And the characteristic of 'training data' may be different by each combination. I wonder the experiment of this study covered all possible sets.
 And the corrupted data is sampled randomly from the other classes. Is there any deviation in the performance by each sampling?

- This study aims to generate the model to be robust to corrupted training data. However, in the result, it is not clear that the proposed method is more robust than others as the AUC/AP from MAW falls (maybe greater than others) as the outlier ratio increases. The authors may give explanations about the result in detail.


Additional Comments
- The readability of Figure 2, 3 is not good. How about showing them on the tables?
- This study shows the superiority from four datasets (image, non-image). However, there is more dataset widely used for novelty detection such as (Fashion) MNIST or MVTech. The authors may consider doing the same experiments on the other dataset.
-The authors may compare the method not only to the novelty detection methods, but many previous works which also aims to be robust to noisy data(or label) in the training process.

---

> ### Author Response · Authors · 2020-11-25
> **Response to Review #3 (part 1)**
>
> Q1. In the description of the proposed method, MAW, Discriminator generates Loss(Lw1) by comparing between Zgen and Zhyp. And Zhyp is unimodal distribution while Zgen follows MoG. I wonder whether there is a risk that inlier and outlier distributions are mixed(combined) as the loss makes the generator generates just the same mu/sigma regardless of the domains. If so, is there any equilibrium trick required so that the generator would not be strong too much?
>
> Response: We don’t think our model prefers a case where both Z’s have the same means. The inliers and outliers should have different structures. If their latent distributions are close, then the set $\cal{Q}$ of distributions q over which we minimize is significantly smaller and this should lead to a larger W1 error and thus larger reconstruction error. We thus believe that the minimization of the reconstruction error avoids this case where the latent distributions of the inliers and outliers are too close.
> Furthermore, in Figure 4 we tested the case where we have a single Gaussian (it is reported in this figure as MAW-single Gaussian and compared among other variants with MAW). Note that the reported performance of the use of a single Gaussian is worse than that of MAW.
>
> Q2. Though it is hard to pre-estimate how the outlier distribution looks like, it is more common to assume the outlier distribution has multi-modal than uni-modal. However, the proposed method approximates the outlier distribution as unimodal Gaussian distribution. Is it possible to model the outliers as multi-modal distribution such as MoG?
>
> Response: While it is possible to model the outliers as a mixture of Gaussians, however, we point out 3 disadvantages of this approach. First of all, the outlier distribution is not important in our work, unlike the inlier distribution. Second of all, the model for MoG of outliers is more complex and the number of mixture components is unclear. Third of all, there is no clear loss for discriminating between the classes of outliers.

---

> ### Author Response · Authors · 2020-11-25
> **Response to Review #3 (part 2)**
>
> Q3. In the experiment with the multiclass dataset, the number of possible inlier domains is the same as the number of classes in the dataset. And the characteristic of 'training data' may be different by each combination. I wonder the experiment of this study covered all possible sets. And the corrupted data is sampled randomly from the other classes. Is there any deviation in the performance by each sampling?
>
> Response: The setup of our experiment is described in detail in Section 4.2 but let us emphasize it here again. We arbitrarily fix one of the classes as the inlier class and randomly sample some inliers from this class and some outliers from the rest of classes. We averaged the experiments over all possible choices of classes for inliers (and respectively for outliers),  percentage of outliers in the test set (the parameter c_test), and three random assignments of inliers and outliers from their respective classes. In Figs. 2 and 3 (of the revised version) we reported such averages (as a point) as well as their corresponding standard deviations (using bars). We believe that we exhausted all possible combinations and even though we averaged (to avoid too many similar figures). Furthermore, the standard deviations are generally not large, so it was okay to consider the different cases in the same figure.
>
> Nevertheless, to support our claim and further address the reviewer’s concern, we performed the following test. For demonstration, we used the COVID-19 dataset and fixed the training outlier ratio = 0.3 and testing outlier ratio = 0.5. We set the inlier class of COVID-19 as the normal chest X-ray images and outlier classes as those which are diagnosed as positive and bacterial pneumonia. We distinguished between two scenarios: fixing inlier samples but randomly sampling outliers, and fixing outlier samples but randomly sampling inliers. For each case, we performed the experiments five times and reported averages and standard deviations (std’s) of their AUCs (Area Under Curve), APs (Average Precision). For the first case (randomly sampled outliers), the averaged AUC is 0.646 and its std is 0.016, also, the averaged AP is 0.523 and its std is 0.015. For the second case (randomly sampled inliers), the averaged AUC is 0.641 and its std is 0.016, also, its averaged AP is 0.523 and its std is 0.017. These results show that MAW’s standard deviations are relatively small and MAW is not sensitive to both types of random sampling.
>
> We further report the experimental results obtained with other choices of inlier classes. We stick to the COVID-19 dataset, which has 3 classes of chest X-ray images: normal, positive COVID-19 and bacterial pneumonia. So there are three different choices for fixing the inliers’ classes. We randomly sample inliers from the corresponding inlier class and also randomly sample the outliers from the rest of the classes. For each case, we also performed the experiments 5 times and reported averages and their stds in the table below.
>
> ========================================================================================
>
> Averaged AUCs, APs and stds when the outlier ratio for training is 0.3 and for testing is 0.5
>
> ++++++++++++++++++++++++++++++++++++++++++++++++++++++++++++++++++++++++++++++++++++++++
>
> Inlier class (the rest is outlier classes) $\hspace{2cm}$ AUC ± std $\hspace{2cm}$ AP ± std
>
> ++++++++++++++++++++++++++++++++++++++++++++++++++++++++++++++++++++++++++++++++++++++++
>
> Normal $\hspace{6.3cm}$ 0.647 ± 0.016 $\hspace{1.4cm}$   0.526 ± 0.017
>
> Positive COVID-19 $\hspace{4.8cm}$ 0.608 ± 0.012  $\hspace{1.4cm}$  0.412 ± 0.012
>
> Bacterial pneumonia $\hspace{4.4cm}$ 0.497 ± 0.020  $\hspace{1.4cm}$  0.318 ± 0.018
>
> ========================================================================================
>
> The standard deviations in the different experiments are small and of similar values, so taking the average does not hide any information e.g., very large standard deviations in one of the instances. However in this example, we have the highest order of variation between the averaged values. Therefore, when we average over the three choices of classes of inliers we lose some information and increase the overall standard deviation, but we believe that this is the best compact way to summarize all possibilities in a fair way. We also remark that averaging over different classes appeared in previous works, for example, Pidhorskyi et al. (2018); Sabokrou et al. (2018).
>
> References:
> Pidhorskyi, Stanislav, Ranya Almohsen, and Gianfranco Doretto. "Generative probabilistic novelty detection with adversarial autoencoders." Advances in neural information processing systems. 2018.
>
> Sabokrou, Mohammad, et al. "Adversarially learned one-class classifier for novelty detection." Proceedings of the IEEE Conference on Computer Vision and Pattern Recognition. 2018.

---

> ### Author Response · Authors · 2020-11-25
> **Response to Review #3 (part 3)**
>
> Q4. This study aims to generate the model to be robust to corrupted training data. However, in the result, it is not clear that the proposed method is more robust than others as the AUC/AP from MAW falls (maybe greater than others) as the outlier ratio increases. The authors may give explanations about the result in detail.
>
> Response: We notice from Figs. 2 and 3 that MAW is generally a top-performing method when considering different datasets even in the very difficult regime of 0.4 and 0.5 outlier corruption ratio of the training set and that other methods also deteriorate in this regime. Also, note that there are some limitations for correctly generating the inlier distribution when it is hard to distinguish between inliers and outliers in the training set. This regime may thus not be the best to consider, but we added it for completeness. Methods that were originally designed for unsupervised anomaly detection (such as RSRAE, IF and LOF), may have an advantage in this regime. We indeed see that for some datasets these latter methods perform slightly better than MAW in this regime, but we don’t see a consistent advantage of one of these methods in this regime when considering all datasets.
>
> One may think that if we use a smaller different choice of $\eta$ (different from our fixed 5/6), then our results may improve. However, we were curious and tested MAW in the regime of 0.4 and 0.5 outlier corruption ratio of the training sets and did not see any significant improvement of accuracy with other choices of $\eta$. We believe that in this region there are obvious limitations on any method that aim to learn characteristics of the inliers, where the inliers are not sufficiently significant.
>
> Q5. The readability of Figure 2, 3 is not good. How about showing them on the tables?
>
> Response: We added the requested tables in Appendix H of our revised version. We still believe that it is easier to grasp information from figures and we thus include the figures in the main text.
>
> Q6. This study shows the superiority from four datasets (image, non-image). However, there is more dataset widely used for novelty detection such as (Fashion) MNIST or MVTech. The authors may consider doing the same experiments on the other dataset.
>
> Response: In order to address your concern, we have extended our experiments to include Fashion MNIST (please check Fig. 3 of the revised version). We thought that we had a sufficient amount of datasets (2 with images and 2 with features), but we are happy to address the challenge and include the suggested dataset.
> The images in MVTech mainly have corruption in a very small region of the whole picture. We believe this dataset is more properly addressed by models that are robust to pixel-wise noise or that consider pixel-level sparsity.
>
> We will eventually include our code on a github page, so practitioners can test it on different datasets.
>
> Q7. The authors may compare the method not only to the novelty detection methods, but many previous works which also aims to be robust to noisy data(or label) in the training process.
>
> Response: Some of the methods we compare with are known to be able to handle noisy data. For instance, the traditional methods (OCSVM, LOF and IF) are robust to “noisy data” and generally can be used in an unsupervised setting (outlier detection). Also, RSRAE is a recent robust method that can be used in the unsupervised setting. DAGMM does not directly address the robustness but its authors tested the effectiveness of their model under a setting where the training dataset is corrupted.
>
> We focused on methods that are directly related to anomaly detection and thus did not find any reason to cite any of the many works on noisy labels or data that are not in this area. We cite RCAE (Chalapathy et al., 2017), but it intends to address noise of a specific structure (e.g. pixelwise sparsity), which is different from our setting (arbitrary corruption) and we thus did not compare with it (our initial experiments were not in favor for RCAE, and we thought it was unfair for this method if we report its results on a different setting).
>
> We could not find a missing method for noisy data that is directly related to our setting and may not be somewhat similar to, or within the same category, of any tested method. If you notice any such method that we should cite, or possibly compare with, please let us know.

---

### Official Review · AnonReviewer4 · 2020-11-01
**This is a solid piece of work, very well written. The problem is well motivated/defined with respect to previous works, which are comprehensively reviewed. The methods and theories are technically sound. Contributions are multi-fold and significant.**

**Rating:** 8
**Confidence:** 3

**Review:**

This paper proposes a robust novelty detection method ("MAW") to model the distribution of the training data in the presence of high fraction (corruption ratios up to 30\%) of outliers. The method add new features to the variational autoencoder (VAE), to detect and isolate the outlier so that the learned distribution only represent the inlier distribution:
1. Uses a carefully designed dimension reduction component to extracts latent lower-dimensional features of the latent distribution.
2. Model the distribution of latent representation as a mixture of Gaussian low-rank inliers and full-rank outliers, both using full covariances instead of diagonal covariances as commonly used in previous VAE-based methods for novelty detection.
3. Penalizes the Wasserstein-1 distance between the data distribution and the latent distribution from the prior distribution. Under a special setting, it theoretically proves that using the Wasserstein-1 metric for regularization yields outliers-robust estimation and is suitable to the low-rank modeling of inliers, while the commonly used Kullback-Leibler (KL) divergence does not.
4. Using the least absolute deviation error for reconstruction.
Experiments on popular anomaly detection datasets demonstrate state-of-the-art results of MAW on standard benchmarks for novelty detection.

================== After rebuttal ==================
Several typos:
In Figure 1, to be consistent the X and Z may be written in lower case.
In Eq. (6), the transpose should be applied to the right $U_1^{(i)}$ instead of the left one.
In Appendix C.1, ``It seems that MAW seems to learn" should be ``It seems that MAW learns" or ``MAW seems to learn".

In Appendix D.3:
In line 2 above Proposition D.2, ``the ill-posedness of (11) with $\mathcal{R}= W_2$": The $\mathcal{R}= W_2$ should be $\mathcal{R}= KL$.
Please rephrase ``the KL divergence fails is unsuitable for low-rank covariance modeling", by e.g., removing ``fails" or inserting ``and" between ``fails is".

---

> ### Author Response · Authors · 2020-11-25
> **Response to Review #4**
>
> We thank and appreciate your encouraging review.

---

### Author Response · Authors · 2020-11-25
**Response to all reviewers with summary of changes**

We thank the reviewers for all their comments; we responded in detail to each reviewer.
We revised our paper accordingly, while using a blue font to indicate the changes. Our main changes are as follows:
(1) We added a paragraph in the introduction that clarifies the common term of ''semi-supervised’’ anomaly detection and distinguishes it from a different use of this term
(2) We added a paragraph in the introduction that clarifies the importance of our new paradigm of learning with a highly corrupted training set. That is, we emphasize two important scenarios, where one may not avoid high corruption of the training set
(3) We added a paragraph in the introduction that emphasizes that in the lack of labels for outliers in the training set, the use of robust losses helps obtain a careful model for the inliers, which is robust to outliers
(4) In our theoretical setting, we replaced the strong constraint $||\mu_1 - \mu_2|| = \epsilon$ with the weaker one $||\mu_1 - \mu_2|| \geq \epsilon$
(5) We extended our numerical experiments to include Fashion MNIST (see Section 4.2)
(6) We extended our numerical experiments to include a different setting, where the training set and the test set are corrupted by outliers with different characteristics (see Section G)
(7) We reported the numerical values of the results depicted in Sections 4.2 and 4.3 in tables (see Section H)

---

### Comment · ~Chieh-Hsin_Lai1 · 2021-01-17
**Response to the final comment of the AC**

Three of the reviewers had wrong claims; we carefully rebutted them, while adding many numerical experiments to address their requests. However, our careful rebuttal was completely ignored by these three reviewers and they did not add any post-rebuttal comments. On the other hand, we thank reviewer 4 (listed first) for the post-rebuttal text, which indicated careful reading of our revisions.

The comment (or meta-review) by the AC has five claims. Three of them were already carefully rebutted by us in our response to the reviewers. The two others are requests to work with an additional dataset and two more benchmarks. However, these were not suggested before the rebuttal period by either the reviewers or the AC. On the other hand, the reviewers proposed other datasets and numerical experiments and we successfully addressed them.

Yours,
Jesse Lai, Dongmian Zou and Gilad Lerman.

---

### Decision · Program_Chairs · 2021-01-07
**Final Decision**

**Decision:**

Reject

**Comment:**

The paper proposes a novelty detection method when training data is itself noisy. A VAE-based approach is developed that promotes robustness of the VAE. The paper assumes that the encoder a two-component Gaussian mixture distribution, individual components denoting inliers and outliers.

The paper hopes that the posterior of the inliers (normal data points) can be represented by a low-rank covariance matrix, while the outliers need a full covariance. Another notable modification is that the Wasserstein-1 regularization is used to replace the KL-regularization in the ELBO, which is claimed to be more suitable to the low-rank modeling.

While this is a relevant problem, and the idea is perhaps interesting, some concerns have been raised.
* The details how to fit the model with the desired mixture posterior in practice is unclear.
* The arguments of section 3 to illustrate the superiority of Wasserstein were found unconvincing, with limiting/unclear assumptions
* The ultra-low latent space dimension (2) is not sufficiently justified
* The experimental section and the selected datasets are small scale, it would be good to include a free larger scale datasets (at least cifar10).
* Comparisons to the open set recognition, or out-of-distribution (OOD) detection would have been a plus.

Overall, this is an OK paper but not yet of sufficient quality.